# TLR4 is a regulator of trained immunity in a murine model of Duchenne muscular dystrophy

Salyan Bhattarai[1,2], Qian Li [1,2], Jun Ding[1,2], Feng Liang[1,2], Ekaterina Gusev[1,2], Orsolya Lapohos [1,2], Gregory J. Fonseca[1,2], Eva Kaufmann [1,2], Maziar Divangahi [1,2,3] & Basil J. Petrof[1,2✉]

Dysregulation of the balance between pro-inflammatory and anti-inflammatory macrophages has a key function in the pathogenesis of Duchenne muscular dystrophy (DMD), a fatal genetic disease. We postulate that an evolutionarily ancient protective mechanism against infection, known as trained immunity, drives pathological inflammation in DMD. Here we show that bone marrow-derived macrophages from a murine model of DMD (mdx) exhibit cardinal features of trained immunity, consisting of transcriptional hyperresponsiveness associated with metabolic and epigenetic remodeling. The hyperresponsive phenotype is transmissible by bone marrow transplantation to previously healthy mice and persists for up to 11 weeks post-transplant. Mechanistically, training is induced by muscle extract in vitro. The functional and epigenetic changes in bone marrow-derived macrophages from dystrophic mice are TLR4-dependent. Adoptive transfer experiments further support the TLR4-dependence of trained macrophages homing to damaged muscles from the bone marrow. Collectively, this suggests that a TLR4-regulated, memory-like capacity of innate immunity induced at the level of the bone marrow promotes dysregulated inflammation in DMD.

[1] Meakins-Christie Laboratories, Translational Research in Respiratory Diseases Program, Research Institute of the McGill University Health Centre, 1001 Decarie Boulevard, EM3-2219, Montreal, QC H4A 3J1, Canada. [2] Department of Medicine, McGill University Health Centre, 1001 Decarie Boulevard, D5, Montreal, QC H4A 3J1, Canada. [3] Department of Microbiology and Immunology, McGill University, Duff Medical Building, 3775 University Street, Montreal, QC H3A 2B4, Canada. ✉email: basil.petrof@mcgill.ca

Duchenne muscular dystrophy (DMD) is one of the most frequent X-linked lethal disorders, affecting approximately 1 in 5000 males[1]. Despite recent advances in cell- and gene-based therapies, DMD remains a devastating disease for which treatment options are extremely limited. The primary cause of the disease is loss of dystrophin[2], a large protein which helps link the internal cytoskeleton to the extracellular matrix of muscle cells, thereby providing mechanical stability to the muscle fiber membrane[3]. Evidence from both animal models and humans indicates that maladaptive inflammatory mechanisms play an integral role in promoting the disease from its earliest stages[4,5].

Macrophages are the most abundant leukocyte population in DMD muscles[6], where their abnormal persistence and dysregulated production of inflammatory mediators play a key role in driving disease progression[7–9]. In previous work, we generated dystrophin-deficient mdx mice (model of DMD) lacking the chemokine receptor CCR2 and demonstrated the importance of bone marrow (monocyte)-derived macrophages (BMDM) in early pathogenesis of the disease[10]. Genetic abrogation of CCR2, which impairs the ability of inflammatory monocytes to exit the bone marrow[11], resulted in reduced numbers of pro-inflammatory macrophages in the dystrophic muscles, accompanied by reduced fibrosis and improved force generation[10]. Intriguingly, genetic ablation of Toll-like receptor 4 (TLR4) in mdx mice led to similar beneficial effects along with a skewing of intramuscular macrophages toward a more anti-inflammatory profile[12]. However, the cellular mechanisms underlying these phenomena are incompletely understood.

It is well established that macrophages have a large capacity to respond and adapt their function to different environmental cues. This plasticity is exemplified by the fact that in vitro stimulation with either interferon-gamma (IFN-γ) or interleukin-4 (IL-4) can polarize the cells toward opposing phenotypes with pro-inflammatory (also called classically activated or M1-like) or anti-inflammatory (alternatively activated or M2-like) profiles, respectively. Moreover, it is well recognized that the above macrophage polarization model greatly oversimplifies and underestimates the true diversity of macrophage phenotypic states[13], particularly during chronic inflammation in vivo. Macrophage phenotype is also modulated by exposure to various pathogen-associated molecular patterns (PAMPs) and endogenous molecules released after tissue injury (damage-associated molecular patterns, or DAMPs), which can serve as ligands for classical receptors of innate immunity such as members of the TLR family[14].

Further complexity is added by the recent recognition that macrophages can undergo epigenetic imprinting to confer a form of innate immune "memory", also known as trained immunity[15–17]. A cardinal feature of trained immunity is that it lacks antigen-specificity and promotes exaggerated cytokine responses to multiple forms of unrelated pathological stimuli[16]. This non-specific hyperresponsiveness of the innate immune system has been mechanistically linked to histone modifications[18–20], as well as alterations in cellular metabolism[21]. Although initially described in the context of infections, there is increasing evidence that trained immunity may also play a deleterious role in different non-infectious diseases associated with chronic sterile inflammation[22–25].

Interestingly, trained immunity can be initiated at the level of myeloid progenitors within the bone marrow[26–29]. This suggests that the process is regulated by systemic factors released from distant sites of inflammation, which can potentially either induce or inhibit trained immunity. Pattern recognition receptors such as TLRs and NOD-like receptors are involved in the regulation of trained immunity in monocytes/macrophages[18,30]. In DMD local immune cell stimulation by DAMPs within the damaged muscles is thought to be a major contributor to disease progression, but the question of whether DAMPs or other factors released from dystrophic muscles might also induce wider epigenetic and functional changes at the level of myeloid cells within the bone marrow, has not been explored.

Here, we show major remodeling of the epigenetic, metabolic and functional inflammatory profiles of monocyte-derived macrophages from the bone marrow of mdx mice. The fact that these changes are found at a distance from the pathological muscle microenvironment implies a role for systemically released mediators such as DAMPs in driving the altered phenotype of these cells. In support of the latter mechanism, we find that abnormal epigenetic and functional characteristics of BMDM from muscular dystrophy mice are TLR4-dependent. Furthermore, the altered phenotype of mdx BMDM demonstrates the hallmark features of trained immunity, including being transmissible and long-lasting after bone marrow transplantation to non-dystrophic mice. Accordingly, trained immunity may be an important mechanism underlying both the generation and maintenance of pathological inflammation in DMD.

## Results

**mdx BMDM exhibit disease stage-specific alterations in basal inflammatory and metabolic profile.** The BMDM from dystrophic mdx mice and age-matched wild-type (WT) controls were studied following 7 days in culture under basal conditions. To determine the influence of disease stage on BMDM phenotype, the cells were harvested at different ages, taking advantage of the fact that skeletal muscle pathology in mdx mice can be separated into prenecrotic (2–3 weeks old), necrotic (6–8 weeks old), and fibrotic (50–60 weeks old) stages of the disease. We first surveyed a panel of classical pro-inflammatory "M1" (iNOS, TNF, IL6, IL12α) and anti-inflammatory "M2" (TGFβ, CD206, Ym1, Arg1) macrophage marker genes in BMDM at the transcriptional level (Fig. 1a–c). During the prenecrotic phase, the expression of these genes was generally equivalent to WT. In contrast, there was significant basal upregulation of multiple pro-inflammatory as well as anti-inflammatory genes in mdx BMDM during the necrotic phase, most of which remained elevated to a lesser extent at the later fibrotic stage. These data indicate that in temporal association with the onset of dystrophic skeletal muscle pathology, the precursor cells in the bone marrow which give rise to macrophages are functionally modified.

To explore potential alterations in inflammatory signaling, we determined the levels of phosphorylated Signal Transducer and Activator of Transcription (STAT) proteins (Fig. 1d–f), which are known to be key players in canonical intracellular pathways linked to macrophage plasticity. Both p-STAT1 and p-STAT3 were unaltered in mdx BMDM during the prenecrotic phase, consistent with the M1 and M2 marker gene expression data. However, at the necrotic stage of disease there was exaggerated phosphorylation of STAT1 and STAT3 in mdx BMDM, which was mitigated but still present during the fibrotic phase. On the other hand, p-STAT6 was not detectable in mdx BMDM during either the necrotic or fibrotic stages (Supplementary Fig. 1).

To investigate whether the changes in mdx BMDM phenotype include signs of metabolic reprogramming, we first assessed mitochondrial oxygen consumption rate (OCR) in BMDM. There were no significant differences in OCR between WT and mdx BMDM during the prenecrotic phase (Fig. 2a). However, significant reductions in both basal and maximal OCR were observed in mdx BMDM during the necrotic phase in comparison to the age-matched WT group (Fig. 2b). In contrast,

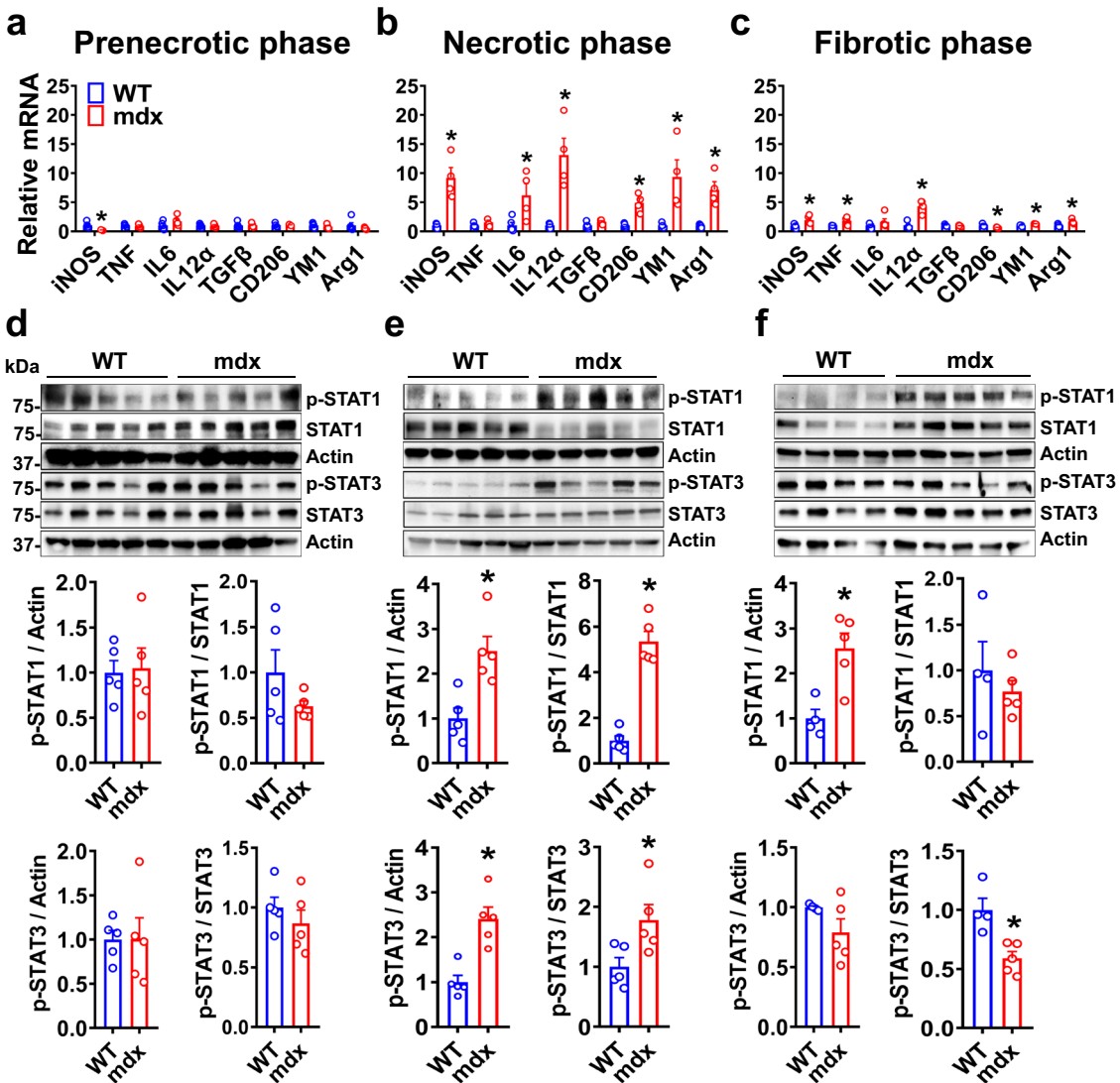

**Fig. 1 Disease stage-dependent alterations in the basal inflammatory status of mdx BMDM. a–c** The basal mRNA transcript levels (expressed relative to the mean age-matched WT level) of prototypical M1 (iNOS, TNF, IL6, and IL12α) and M2 (TGFβ, CD206, YM1, and Arg1) marker genes during the: **a** prenecrotic ($n = 4$ for iNOS, TNF, IL12α, Arg1 genes in mdx; rest $n = 5$/group), **b** necrotic ($n = 5$ for WT, $n = 4$ for mdx), and **c** fibrotic phases ($n = 4$ for WT, $n = 5$ for mdx) of disease, are shown. The relative levels of phospho-STAT1, total STAT1, phospho-STAT3 and total STAT3 in cell lysates of WT and mdx BMDM from the same disease phases (**d–f**, respectively; all experimental replicates are shown) are also demonstrated; **d** ($n = 5$/group), **e** ($n = 5$/ group), **f** ($n = 4$ for WT, $n = 5$ for mdx). Data represent means ± SEM of biologically independent samples from different mice. *$P < 0.05$ vs. WT (unpaired $t$-test, two-tailed). See Source Data file for the exact $P$-values.

mdx BMDM from the fibrotic phase showed a reversal of this pattern with increased maximal OCR relative to age-matched WT values (Fig. 2c). Along the same lines, there were no significant differences in culture media lactate concentrations (index of glycolysis) between WT and mdx BMDM during the prenecrotic phase, whereas in older mice lactate was increased during the necrotic phase and decreased during the fibrotic phase in the mdx groups (Fig. 2d). Interestingly, in WT mice the induction of massive muscle necrosis by bilateral hindlimb muscle (tibialis anterior) injection of the myotoxic agent cardiotoxin in vivo did not trigger comparable alterations in either the inflammatory (Fig. 2e) or metabolic (Fig. 2f) profiles of BMDM. Repeated bouts of acute cardiotoxin-induced muscle damage were similarly unable to recapitulate the BMDM phenotype of mdx mice (Fig. 2e). These results imply that chronic ongoing muscle injury is required to induce the abnormal immunometabolic phenotype observed in mdx BMDM.

**mdx BMDM respond in an exaggerated fashion to heterologous inflammatory stimuli.** We wished to determine whether mdx BMDM exhibit non-specific amplification of gene transcription responses after exposure to heterologous stimuli, which is a defining feature of trained immunity. For this purpose, we employed a variety of unrelated secondary stimuli (cytokines, PAMPs/DAMPs). Following acute exposure to the classical M1-polarizing stimulus of LPS + IFN-γ, mdx BMDM from the necrotic phase showed increased pro-inflammatory gene (iNOS, TNF, IL6, IL12α) transcript upregulation compared to WT BMDM (Fig. 3a). Greater anti-inflammatory gene expression (TGFβ, CD206, Ym1, Arg1) was similarly observed in mdx BMDM after exposure to the M2-polarizing cytokine IL-4 (Fig. 3b). Stimulation with fibrinogen, an endogenous TLR4 ligand DAMP previously implicated in DMD pathogenesis[31], also triggered greater pro-inflammatory (Fig. 3c) as well as anti-inflammatory (Fig. 3d) gene transcript levels in the mdx BMDM

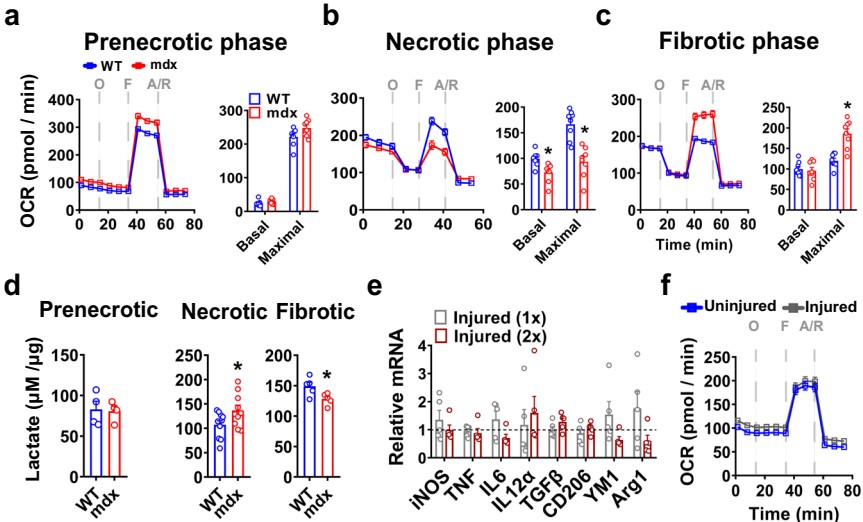

**Fig. 2 Metabolic changes in mdx BMDM at different disease stages. a–c** Oxygen consumption rate (OCR) values after sequential treatment of BMDM with oligomycin (O), FCCP (F) and antimycin + rotenone (A/R) as measured by the Seahorse XF Analyzer at: **a** prenecrotic ($n = 8$/group), **b** necrotic ($n = 8$/group), and **c** fibrotic ($n = 8$/group) phases of disease. OCR values are normalized to cell count. **d** Lactate levels (normalized to μg of cell lysate protein) in supernatants from WT and mdx BMDM after incubating the cells for 48 h in phenol-red-free RPMI culture media ($n = 4$/group for prenecrotic, $n = 10$/group for necrotic, and $n = 5$/group for fibrotic). **e** mRNA transcript levels in BMDM (expressed relative to the mean uninjured level represented by the dashed line) obtained from WT mice after acute muscle injury (Injury 1x = injection of cardiotoxin into both tibialis anterior muscles with euthanasia 3 days later; Injury 2x = injection of cardiotoxin into tibialis anterior and gastrocnemius muscles on days 0 and 3, respectively, with euthanasia at day 13) ($n = 5$/group). **f** Representative OCR curve from Injury 1x group in **e** ($n = 5$/group). Data represent means ± SEM of biologically independent samples from different mice. *$P < 0.05$ vs. WT (unpaired $t$-test, two-tailed). See Source Data file for the exact $P$-values.

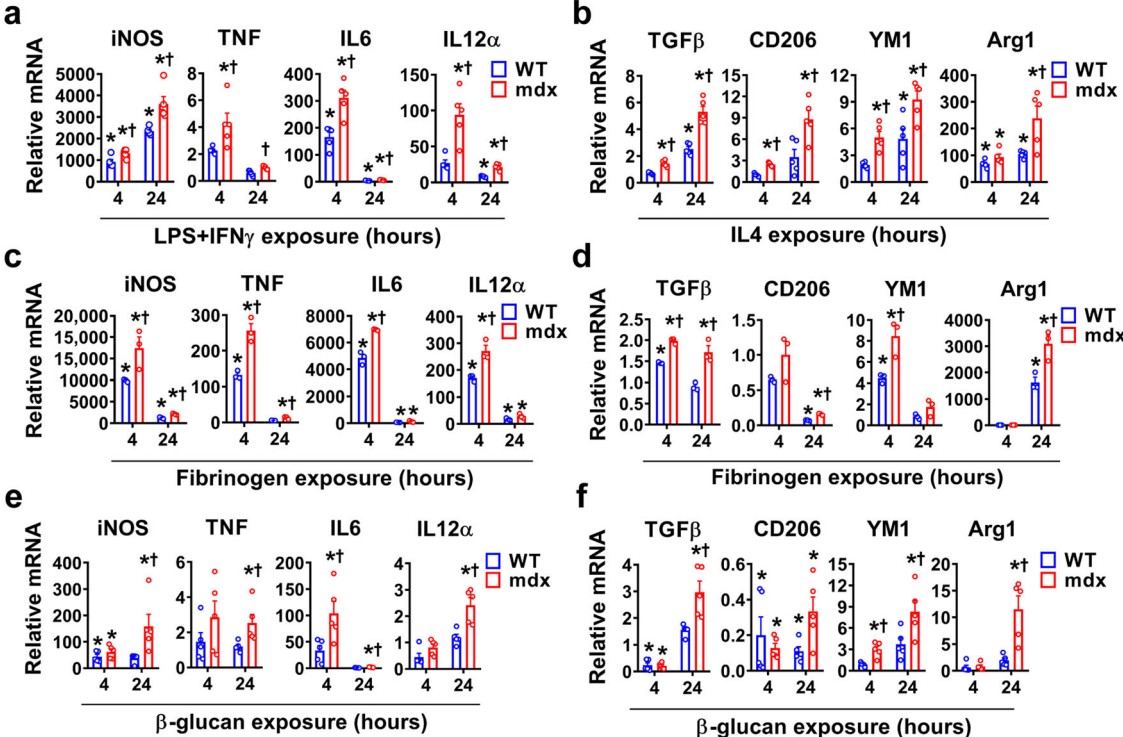

**Fig. 3 mdx BMDM exhibit non-specific amplification of M1 and M2 marker gene expression in response to heterologous stimuli. a–f** BMDM from age-matched WT and mdx mice at the necrotic phase of disease were exposed to: **a** LPS + IFNγ ($n = 5$/group), **b** IL4 ($n = 5$/group), **c, d** fibrinogen ($n = 3$/group), or **e, f** β-glucan for 4 and 24 h ($n = 5$/group); mRNA transcript data for prototypical M1 and M2 marker genes are expressed relative to the mean basal WT (unstimulated) level determined on the same PCR plate. Data represent means ± SEM of biologically independent samples from different mice. *$P < 0.05$ vs. unstimulated WT and †$P < 0.05$ vs. stimulated WT at a given time point (one-way ANOVA followed by Tukey post-hoc test, two-tailed). See Source Data file for the exact $P$-values.

group. Furthermore, we exposed BMDM to the fungal cell wall-derived PAMP β-glucan, as a non-specific secondary stimulus which is entirely unrelated to DMD pathogenesis. Once again, significant hyperresponsiveness was observed in mdx BMDM relative to the WT group for β-glucan (Fig. 3e, f). An analogous but less pronounced pattern of generalized hyperresponsiveness to these stimuli was also found in mdx BMDM from the fibrotic phase of the disease (Supplementary Fig. 2). On the other hand, BMDM from prenecrotic mdx mice (Supplementary Fig. 3a–c) as well as cardiotoxin-injured mice (Supplementary Fig. 3d, e) failed to exhibit similarly amplified responses to secondary stimuli in comparison to WT BMDM. These data show that following the onset of chronic generalized skeletal muscle necrosis in mdx mice, BMDM are profoundly altered and demonstrate significant hyperresponsiveness to multiple forms of unrelated inflammatory stimuli.

**Sustained phenotypic reprogramming of mdx BMDM.** To account for the altered BMDM phenotype observed in post-necrotic mdx mice, we postulated that DAMPs released into the systemic circulation from widespread muscle damage could be at least partly responsible for these changes. To test this hypothesis, naive WT BMDM were first exposed to crushed skeletal muscle extract (ME) derived from either WT or mdx mice to act as a primary training stimulus; the ME was removed after 24 h, and the BMDM were then exposed 5 days later to fibrinogen acting as a secondary stimulus (Fig. 4a). For the majority of genes showing enhanced upregulation after the first challenge with ME, expression had largely returned to baseline levels prior to the secondary challenge with fibrinogen (Supplementary Fig. 4a, b). In comparison to the control BMDM (PBS group), the BMDM exposed to either WT-ME or mdx-ME demonstrated greater pro-inflammatory (Fig. 4b) as well as anti-inflammatory (Fig. 4c) gene upregulation in response to fibrinogen. Thus, exposing naive WT BMDM to components of damaged skeletal muscle induced a phenotype resembling the mdx BMDM. The magnitude of this effect was dependent on both the concentration (Supplementary Fig. 4c) and duration (Supplementary Fig. 4d) of exposure to ME, which could account for the fact that mdx serum induced lesser effects than the highest dose of ME (Supplementary Fig. 4e). Overall, these findings support the concept that muscle damage-related molecules could potentially act as a primary "training stimulus" for the induction of trained immunity in BMDM of mdx mice.

To build on the above in vitro evidence for trained immunity, we next performed in vivo experiments whereby irradiated WT host mice received bone marrow transplants from either WT (WT→WT) or mdx (mdx→WT) donor mice (Fig. 4d). At 11 weeks after bone marrow reconstitution, there was no pathological macrophage infiltration in muscles of the WT host mice that received mdx donor BMDM (Supplementary Fig. 4f). However, the significant functional differences between WT and mdx origin BMDM remained intact when these cells were studied in vitro. Hence mdx origin BMDM continued to show greater pro- and anti-inflammatory gene responses after stimulation with fibrinogen (Fig. 4e, f) or β-glucan (Fig. 4g, h) despite being transplanted into WT mice. This in vivo transmissibility and long-term maintenance of the hyperresponsive mdx BMDM phenotype within a non-dystrophic WT host environment is consistent with the presence of innate immune memory[29].

**Phenotypic reprogramming of mdx BMDM is TLR4-dependent.** TLR4 has the ability to sense a wide range of DAMPs released after tissue injury. Therefore, we generated mdx mice with genetic abrogation of TLR4 (mdxTLR4$^{-/-}$) to determine the role of TLR4 in the altered immunophenotype of mdx BMDM. Under basal conditions mdxTLR4$^{-/-}$ BMDM exhibited lower transcript levels of both pro- and anti-inflammatory genes compared to mdx BMDM, most strikingly during the necrotic stage of the disease (Fig. 5a). In this regard, unsupervised hierarchical clustering (Fig. 5b) clearly separated mdx from WT and mdxTLR4$^{-/-}$ BMDM during the necrotic phase; this genotype clustering was less pronounced during the fibrotic phase and completely absent at the prenecrotic stage. In addition, mdxTLR4$^{-/-}$ BMDM from both the necrotic (Fig. 5c) and fibrotic (Fig. 5d) stages exhibited a higher maximal OCR level compared to their age-matched mdx BMDM counterparts. Exposure to unrelated secondary stimuli (Fig. 5e–h) revealed that the expression of both pro-inflammatory and anti-inflammatory genes was reduced in mdxTLR4$^{-/-}$ BMDM in comparison to necrotic phase mdx BMDM. The changes in mRNA transcript levels were corroborated at the protein level by western blotting (Fig. 5i) and ELISA (Fig. 5j). Furthermore, using the same in vitro ME stimulation protocol which induced signs of trained immunity in WT BMDM (Fig. 4a), we found that this training response to ME was prevented in cultured BMDM from TLR4-deficient (non-dystrophic) mice. This was evinced by absence of the characteristic amplified gene expression responses to fibrinogen and β-glucan (Fig. 5k, l). Overall, these data indicate that the hyperresponsiveness to unrelated secondary inflammatory stimuli as well as the altered metabolic phenotype observed in mdx BMDM are largely abrogated by TLR4 deficiency.

**mdx BMDM are epigenetically altered in a TLR4-dependent manner.** Epigenetic reprogramming is an important mechanism underlying the development of trained immunity. We performed chromatin immunoprecipitation (ChIP)-seq to quantify tri-methylation of histone 3 lysine 27 (H3K27me3), a histone mark classically associated with gene silencing[32]. Genome-wide analysis revealed a significant reduction of H3K27me3 signal intensity in mdx BMDM compared to WT BMDM; in contrast the mdxTLR4$^{-/-}$ BMDM showed an overall increase in H3K27me3 signal relative to WT (Fig. 6a). The three experimental groups were next analyzed according to their different patterns of relative H3K27me3 signal intensity in close proximity (−5 kb to +1 kb) to the gene body across the genome (see Supplementary Data 1 table). This analysis identified four distinct patterns (termed GP1–GP4) showing dynamic regulation in the mdx group relative to WT (Fig. 6b). Among the mdx-altered H3K27me3 peaks, by far the predominant pattern was a relative reduction of the signal in mdx compared to WT, with restoration to WT levels in the mdxTLR4$^{-/-}$ group (GP1 = 69% of peaks). This pattern was followed in frequency by a reduction in mdx without restoration in mdxTLR4$^{-/-}$ (GP2 = 24% of peaks); increase in mdx without restoration in mdxTLR4$^{-/-}$ (GP3 = 5% of peaks); and increase in mdx with restoration in mdxTLR4$^{-/-}$ (GP4 = 2% of peaks).

Pathway analysis was next applied to the mdx-altered H3K27me3 signal patterns using Reactome informatics (Supplementary Data 2 table). For the largely predominant GP1 pattern, extracellular matrix- and collagen-related processes were among the most significantly enriched biological pathways (Fig. 6c; see Supplementary Fig. 5 for GP2-4 patterns). Individual genes selected a priori to reflect key pro-inflammatory, anti-inflammatory, and pro-fibrotic processes were also examined (Supplementary Data 3 table). In comparison to the WT group, there was a widespread decrease of the H3K27me3 mark in mdx BMDM for all of these gene categories, which was significantly reversed in the mdxTLR4$^{-/-}$ group (Fig. 6d, e). Dynamically regulated patterns of the H3K27me3 mark in mdx BMDM were also used to perform

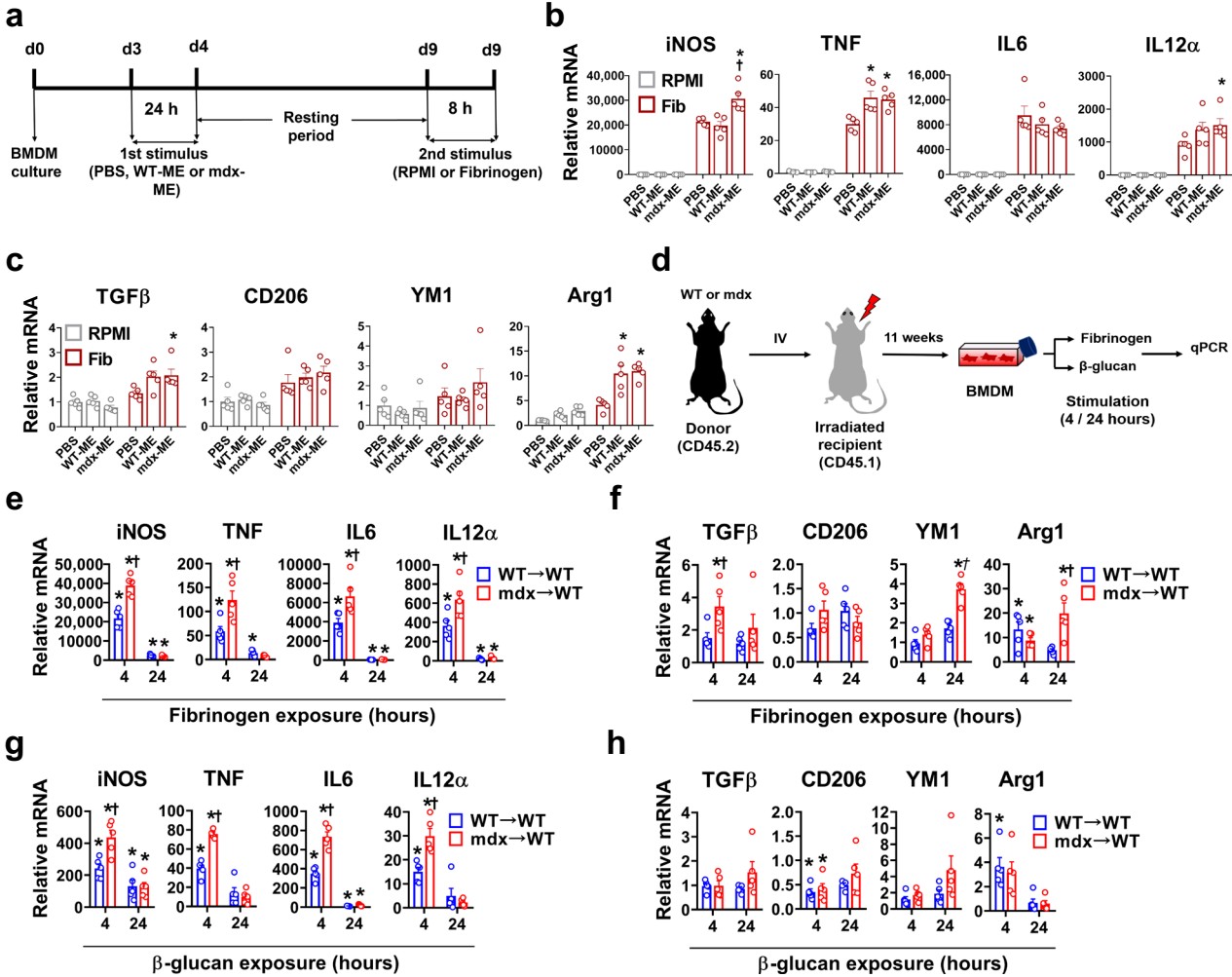

**Fig. 4 BMDM exposed to muscle damage show a trained-like phenotype both in vitro and in vivo. a** Experimental design for WT- or mdx- muscle extract (ME) exposure to induce trained immunity in naive WT BMDM. **b, c** Transcript levels are shown for BMDM trained with PBS, WT-ME, or mdx-ME and then secondarily exposed to fibrinogen (Fib) or RPMI for 8 h; the mRNA levels determined by qPCR are expressed relative to the mean control (PBS-trained and RPMI-stimulated) value ($n = 5$/group). **d** Schematic representation of the BM transplant chimeric model. At 11 weeks post-transplantation BMDM were generated from recipient mice previously transplanted with whole BM from either WT (WT → WT) or mdx (mdx → WT) mice at necrotic phase. The BMDM were stimulated with fibrinogen or β-glucan for 4 and 24 h (**e–h**); **e** ($n = 4$ for TNF in 24 h mdx group, rest $n = 5$/group), **f** ($n = 5$/group), **g** ($n = 4$ for TNF in 4 h mdx group, rest $n = 5$/group), **h** ($n = 4$ for TGFβ and CD206 WT 4 h groups, rest $n = 5$/group). Data represent means ± SEM of biologically independent samples from different mice. **b, c** *$P < 0.05$ vs. PBS-trained and †$P < 0.05$ vs. WT-ME trained. **e–h** *$P < 0.05$ vs. WT-unstimulated and †$P < 0.05$ vs. WT-stimulated group at a given time point (one-way ANOVA followed by a Tukey post-hoc test, two-tailed). See Source Data file for the exact $P$-values.

transcription factor-related pathway analysis (Supplementary Fig. 6). The latter analysis indicated that for the most frequently observed H3K27me3 pattern (termed PP1), transcription factor enrichment was found for genes linked to the VENTX homeobox protein, RUNX3/Wnt signaling, and several interleukins (IL-21, IL-35, IL-23, IL-20, IL-12, IL-2) (Supplementary Fig. 6; Supplementary Data 4 and 5 tables).

ChIP-qPCR was employed to corroborate these results by quantifying the H3K27me3 mark within promoter regions of the prototypical pro- and anti-inflammatory marker genes. In keeping with the ChIP-seq data, the mdx group showed lower H3K27me3 promoter occupancy than in WT (Fig. 6f), whereas levels of this histone mark were relatively increased in the mdx BMDM lacking TLR4 (Fig. 6g). ChIP-PCR was additionally utilized to assess promoter occupancy by H3K4me3 (Supplementary Fig. 7), a permissive histone mark typically associated with gene activation. Greater H3K4me3 promoter occupancy was observed for several genes in the mdx group (Supplementary

Fig. 7a), but in contrast to H3K27me3, genetic abrogation of TLR4 in mdx BMDM had no significant impact on H3K4me3 deposition (Supplementary Fig. 7b).

In addition to changes in methylation, increased acetylation (ac) of H3K27 is linked to greater chromatin accessibility and increased transcriptional competency. ChIP-seq revealed that in comparison to WT BMDM, there was an overall reduction of H3K27ac signal intensity in mdx BMDM, with a further decrease in the mdxTLR4$^{-/-}$ group (Supplementary Fig. 8a, b; Supplementary Data 1 table). Reactome pathway analysis (Supplementary Data 2 table) of this dominant pattern (GP1 = 73% of peaks) indicated particular enrichment for biological processes associated with protein metabolism, the cell cycle, and gene expression (Supplementary Fig. 8c). The next most frequent pattern was increased H3K27ac signal in mdx BMDM compared to WT and a relative decrease in the mdxTLR4$^{-/-}$ group (GP2 = 16% of peaks). Interestingly, this pattern was especially enriched for cellular responses to stress and external stimuli

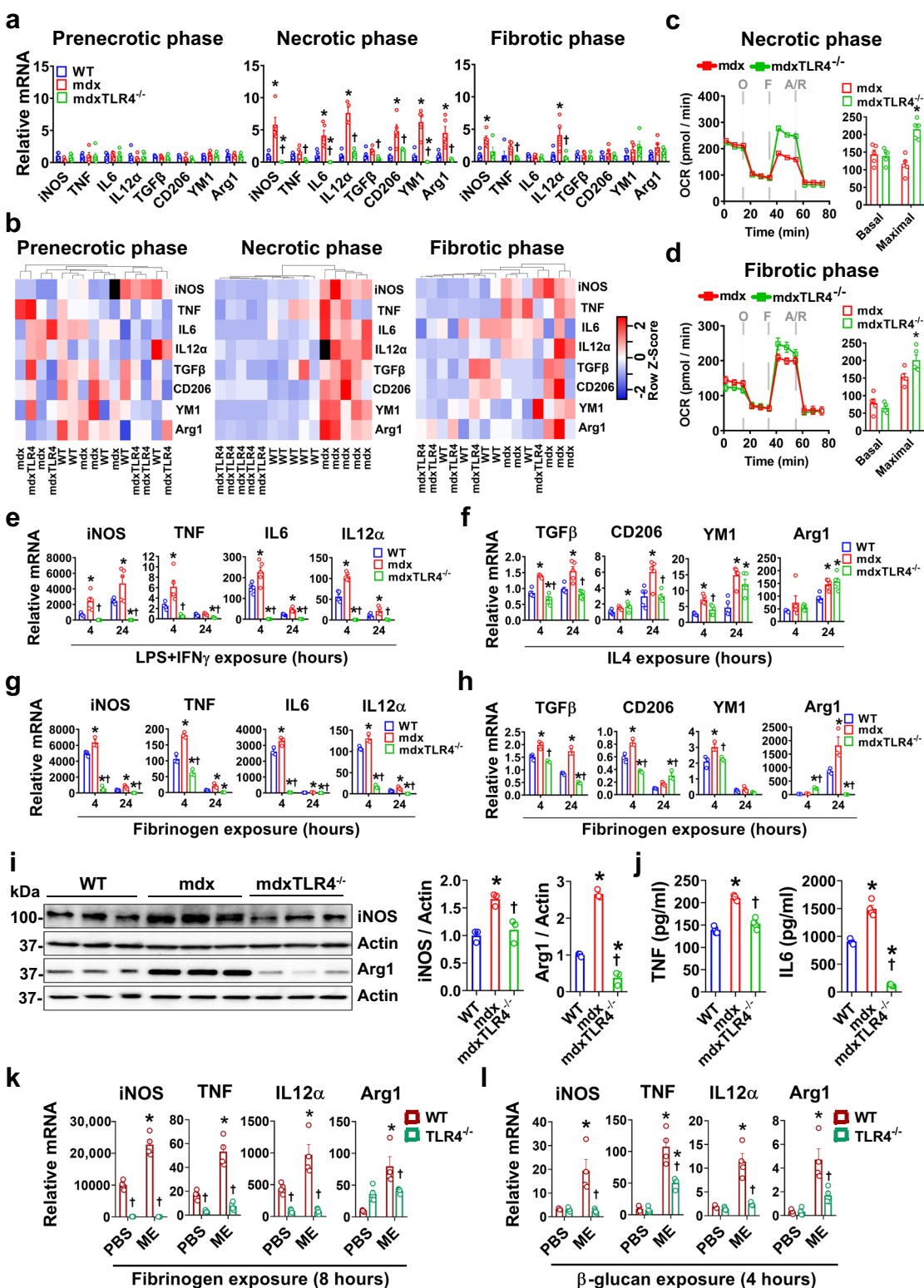

(Supplementary Fig. 8d), which is consistent with the trained immunity phenotype. Transcription factor-related pathway analysis was also performed (Supplementary Fig. 9; Supplementary Data 4 and 5 tables), and the corresponding peak patterns suggested possible involvement of interleukin, Notch, and interferon signaling.

Taken together, these data suggest a complex pattern of epigenetic reprogramming in mdx BMDM, with co-existence of

histone mark modifications that can either augment (decreased H3K27me3 and increased H3K4me3) or reduce (decreased H3K27ac) the open chromatin state. The predominant histone mark changes demonstrated in mdxTLR4$^{-/-}$ BMDM, on the other hand, would generally be expected to decrease chromatin accessibility (increased H3K27me3 and decreased H3K27ac). In addition, among the epigenetic modifications observed in mdx BMDM, it was most clearly evident for H3K27me3 that the changes

**Fig. 5 TLR4 regulates the altered phenotype of mdx BMDM. a** Basal M1 and M2 marker gene transcript levels (expressed relative to the mean age-matched WT level) in BMDM from WT, mdx, mdxTLR4$^{-/-}$ mice at different phases of disease ($n = 4$ for iNOS in prenecrotic phase and IL12α in necrotic phase mdx groups, rest $n = 5$/group). **b** Heatmap showing unsupervised hierarchical clustering analysis of samples based on the gene expression data shown in **a**. Genes with higher and lower expression levels are identified as red or blue (variable units proportional to color intensity), respectively, whereas outlier samples are black. **c, d** Oxygen consumption rate (OCR) data for mdx and mdxTLR4$^{-/-}$ BMDM at necrotic ($n = 5$/group) and fibrotic phases ($n = 5$/group) of the disease. **e–h** WT, mdx, and mdxTLR4$^{-/-}$ BMDM at necrotic phase were stimulated with LPS + IFNγ ($n = 5$/group) or IL4 ($n = 5$/group) or fibrinogen ($n = 3$/group) to determine mRNA transcript levels (expressed relative to the mean WT-unstimulated value) of: (**e, g**) M1 and (**f, h**) M2 marker genes. **i** western blot images and densitometric analyses of total iNOS and Arginase1 in cell lysates of fibrinogen-stimulated (24 h) WT, mdx and mdxTLR4$^{-/-}$ BMDM from necrotic phase mice ($n = 3$/group; all experimental replicates are shown). **j** ELISA measurements of TNF and IL6 protein levels in culture supernatants from WT, mdx and mdxTLR4$^{-/-}$ BMDM after stimulation with fibrinogen for 24 h ($n = 4$/group). **k, l** As shown in Fig. 4a, WT and TLR4$^{-/-}$ BMDM were trained with mdx-ME and secondarily challenged with **k** fibrinogen (8 h; $n = 4$/group) or **l** β-glucan (4 h; $n = 4$/group), followed by qPCR. The graph shows mRNA transcript levels of genes relative to the mean control (PBS-trained and RPMI-stimulated) WT group. Data represent means ± SEM of biologically independent samples from different mice; **a** *$P < 0.05$ WT vs. mdx and †$P < 0.05$ mdx vs. mdxTLR4$^{-/-}$ (one-way ANOVA followed by Tukey post-hoc test, two-tailed); **c, d** *$P < 0.05$ WT vs. mdx (unpaired $t$-test, two-tailed); **e–j** *$P < 0.05$ vs. WT and †$P < 0.05$ vs. mdx at a given time point (one-way ANOVA followed by a Tukey post-hoc test, two-tailed); **k** *$P < 0.05$ PBS vs. ME and †$P < 0.05$ WT vs. TLR4$^{-/-}$ group (two-way ANOVA followed by a Tukey post-hoc test, two-tailed). See Source Data file for the exact $P$-values.

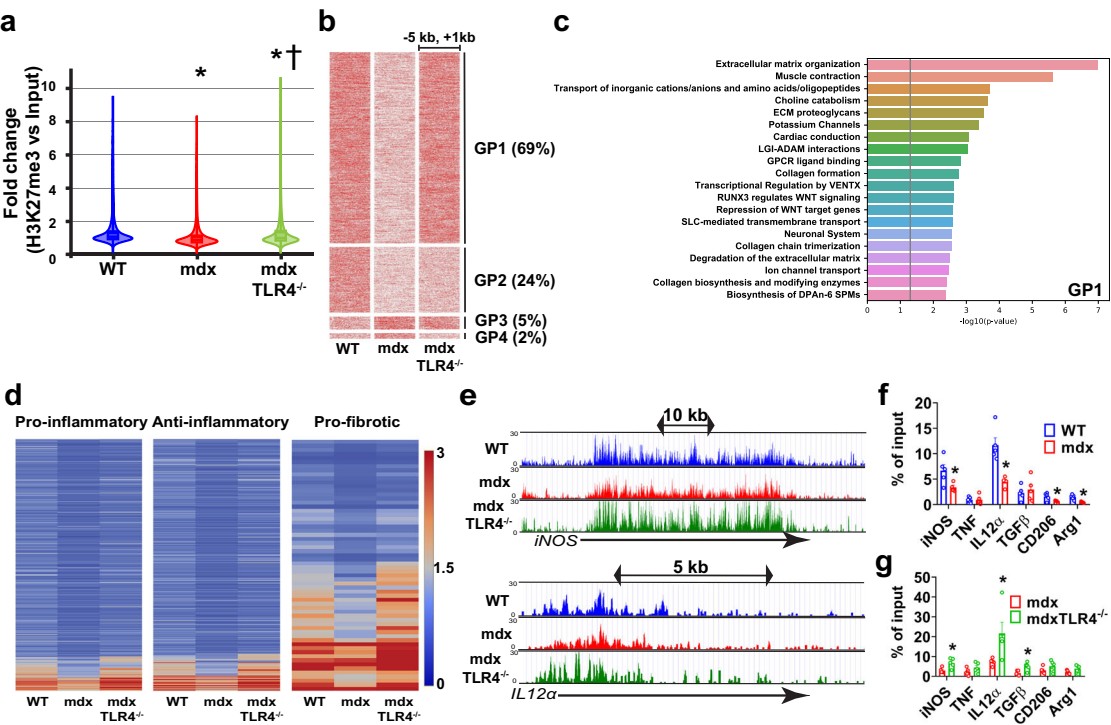

**Fig. 6 Dynamic regulation of H3K27me3 is present in mdx BMDM and mediated by TLR4.** ChIP sequencing was performed in WT, mdx and mdxTLR4$^{-/-}$ BMDM: **a** Violin plots representing normalized intensity fold-change (H3K27me3 vs. Input) in the −5 kb to +1 kb neighboring genomic region around all genes across the whole genome; *$P < 0.0001$ compared to WT BMDM and †$P < 0.0001$ compared to mdx group (two-sided Mann–Whitney $U$-test). **b** Heatmaps showing, in order of frequency (indicated by the percentages in parentheses), the different Gene-based Patterns (GP1-4) of the normalized H3K27me3 read intensity (variable units proportional to red color intensity) within the nearby region [−5 kb to +1 kb] of the gene body for WT, mdx and mdxTLR4$^{-/-}$ BMDM. **c** Pathway enrichment analysis (Reactome database) for genes showing the predominant GP1 configuration (Supplementary Data 2 table shows the extended list for all Gene-based Patterns). The vertical line indicates the cutoff $P$-value = 0.05. **d** Heatmaps of H3K27me3 intensity for pre-defined genes representing prototypical pro-inflammatory, anti-inflammatory, and pro-fibrotic pathways (see Supplementary Data 3 table for complete gene lists). **e** Representative pro-inflammatory gene loci (iNOS, IL12α) showing the lower H3K27me3 peak intensity in the mdx group compared to WT and mdxTLR4$^{-/-}$. **f, g** ChIP-qPCR was performed to detect H3K27me3 occupancy on the promoters of M1 and M2 marker genes in: **f** WT versus mdx BMDM ($n = 4$ for iNOS, IL12α, CD206 and Arg1 gene in mdx group, rest $n = 5$/group) and **g** mdx versus mdxTLR4$^{-/-}$ BMDM ($n = 5$/group); IgG was used as a control for non-specific binding of antibody. Data represent means ± SEM of biologically independent samples from different mice. *$P < 0.05$ by unpaired $t$-test, two-tailed). See Source Data file for the exact $P$-values.

were TLR4-dependent and could promote increased transcriptional activation of genes involved in inflammation and fibrosis.

**Functional differences between WT, mdx, and mdxTLR4$^{-/-}$ BMDM in vivo.** Finally, to determine whether the changes in

mdx BMDM phenotype in vitro are associated with functional differences in vivo, we performed intravenous (IV) adoptive transfer of bone marrow from necrotic phase mdx mice or age-matched WT mice (bone marrow donors) into WT recipient mice (hosts) subjected to acute hindlimb muscle injury via cardiotoxin injection (Fig. 7a). The bone marrow donor mice shared the

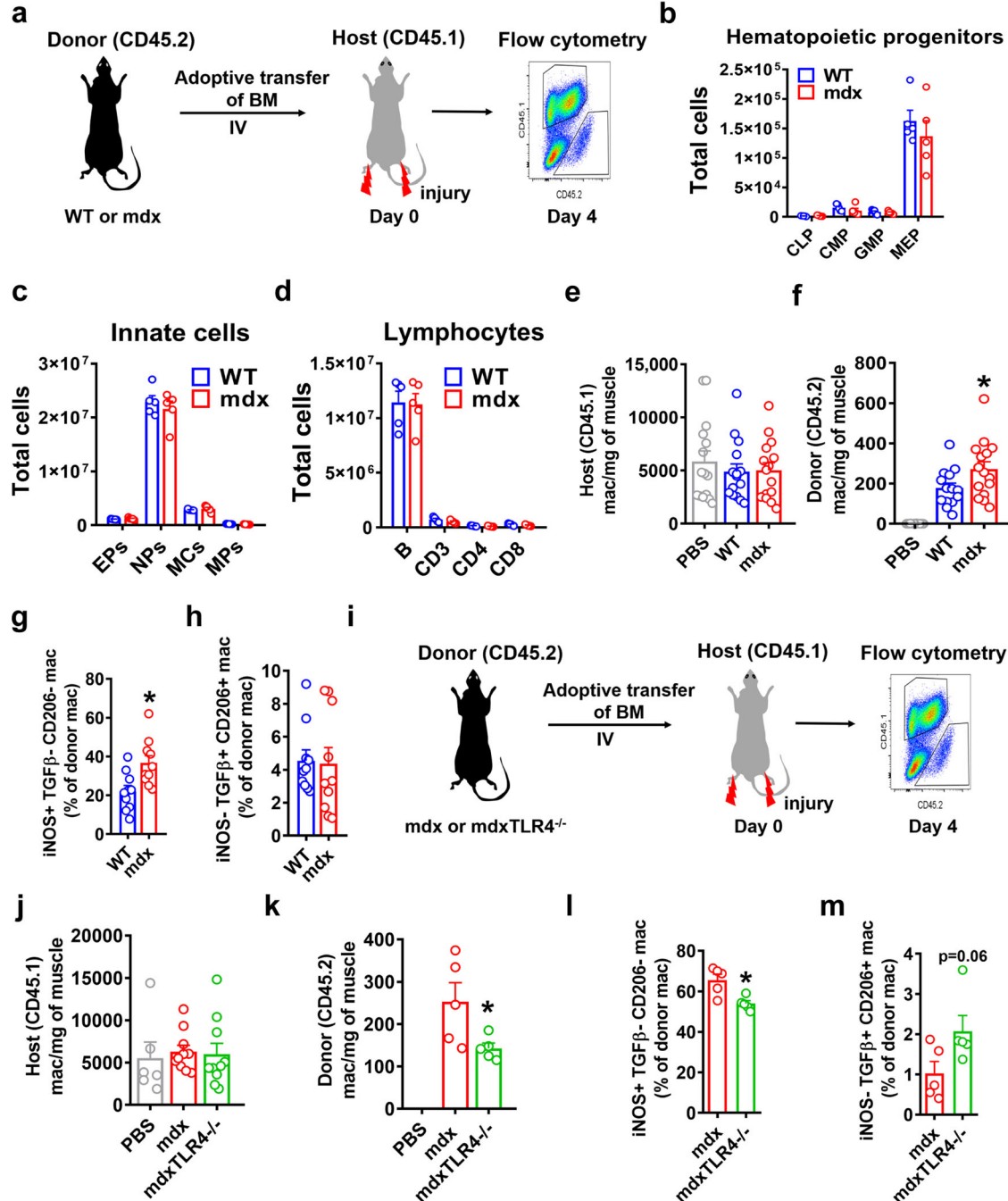

**Fig. 7 Adoptive transfer reveals the altered phenotype of mdx and mdxTLR4⁻/⁻ BMDM after muscle injury in vivo. a** Pictorial representation showing adoptive transfer of bone marrow (BM) from WT or mdx (both CD45.2 allele) mice into non-irradiated congenic WT (CD45.1 allele) mice at the onset of cardiotoxin-induced hindlimb muscle injury, followed by flow cytometric analysis of donor and host BMDM in the injured muscle 4 days later. **b–d** Comparison of absolute number of: **b** Hematopoietic precursor cells (CLP: common lymphoid progenitors, CMP: common myeloid progenitors, GMP: granulocyte/macrophage progenitors, MEP: megakaryocyte/erythrocyte progenitors; $n = 5$/group), **c** Innate myeloid cells (EPs: Eosinophils, NPs: Neutrophils, MCs: Monocytes, MPs: Macrophages) ($n = 5$/group) and **d** Lymphocytes (B: B cells, CD3: CD3 T cells, CD4: CD4 T cells, CD8: CD8 T cells) ($n = 5$/group) in the BM of age-matched WT or mdx at necrotic phase. **e, f** Quantification of host (CD45.1–WT; **e** $n = 15$/group) and donor (CD45.2-either WT or mdx or mock adoptive transfer with PBS; **f** $n = 14$ for WT, rest $n = 15$/group) macrophages in the injured muscles (defined as CD45 + CD11c-CD11b + F4/80 + ). **g** Percentage of donor pro-inflammatory (iNOS + TGFβ-CD206-) and **h** donor anti-inflammatory (iNOS-TGFβ + CD206 + ) BMDM of either WT or mdx origin in the injured WT host muscle ($n = 10$/group). **i** Schematic showing adoptive transfer of mdx or mdxTLR4⁻/⁻ BM using the same experimental design. **j, k** Quantification of host (CD45.1–WT; $n = 6$ for PBS, rest $n = 10$/group) and donor (CD45.2-either mdx or mdxTLR4⁻/⁻; $n = 5$/group) macrophages in the injured muscles. **l** Percentage of donor pro-inflammatory and **m** donor anti-inflammatory BMDM of either mdx or mdxTLR4⁻/⁻ origin in the injured WT host muscle ($n = 5$/group). Data represent means ± SEM of biologically independent samples from different mice. *$P < 0.05$ (unpaired $t$-test, two-tailed). See Source Data file for the exact $P$-values.

common CD45.2 allele and both strains (WT and mdx) demonstrated an equivalent distribution of bone marrow leukocyte populations prior to adoptive transfer (Fig. 7b–d). The bone marrow recipient WT host mice had the CD45.1 allele in order to allow donor versus host origin macrophages to be distinguished by flow cytometry. Muscles were collected from the WT host mice at 4 days post-injury, which was the approximate time of maximal macrophage infiltration.

The majority of macrophages in the injured muscle were of host origin (CD45.1), and there were no differences in CD45.1 macrophage number between the WT and mdx donor groups (Fig. 7e). However, donor origin (CD45.2) BMDM in the injured muscles were significantly more abundant in the mice injected with mdx donor cells as compared to WT donors (Fig. 7f). As muscles of mdx mice from the necrotic phase of disease contain a higher proportion of pro-inflammatory (iNOS + , TGFb-, CD206-) macrophages (Supplementary Fig. 10), we also sought this phenotypic signature in the adoptively transferred BMDM. In the host mice that received mdx bone marrow donor cells, the injured muscles contained a greater percentage of CD45.2 macrophages exhibiting a pro-inflammatory phenotype in comparison to mice which had received WT donor cells (Fig. 7g, h). Importantly, these differences in inflammatory profile were limited to the CD45.2 (donor) macrophages and not observed in CD45.1 (host) macrophages (Supplementary Fig. 11a, b) within the same muscles.

The same experimental design was used to compare the in vivo behavior of mdx versus mdxTLR4$^{-/-}$ bone marrow origin macrophages (Fig. 7i). Once again there were no differences in the numbers (Fig. 7j) or inflammatory profile (Supplementary Fig. 11c, d) of host origin (CD45.1) macrophages within the injured WT muscles. In contrast, adoptively transferred (CD45.2) macrophages were reduced in number (Fig. 7k) and also less inflammatory (Fig. 7l, m) in the injured WT muscles of mice that received mdxTLR4$^{-/-}$ donor cells as compared to the group injected with mdx donor cells. These in vivo adoptive transfer studies are in keeping with the earlier in vitro data and support the presence of an altered macrophage phenotype in mdx mice that is epigenetically imprinted at the level of the bone marrow and significantly mediated via TLR4.

## Discussion

From an evolutionary standpoint, the immune system is primarily designed to deal with acute localized muscle damage (e.g., traumatic or infectious) rather than the ongoing widespread injury found in chronic muscle diseases such as DMD. After acute muscle injury an initial wave of bone marrow (monocyte)-derived M1-like macrophages populates the muscle, which are then replaced by macrophages with a M2-like phenotype[33]. As a general rule, any interference with this tightly regulated sequence of events impedes normal skeletal muscle repair[34]. Importantly, this normal sequence is lost in the context of DMD, where repeated injury leads not only to an abnormal persistence of macrophages, but also to an increased prevalence of macrophages within the muscle that exhibit simultaneous upregulation of both M1 and M2 marker genes[7–9]. Failed muscle regeneration and subsequent fibrosis in dystrophin deficiency have been substantially attributed to dysregulated function of macrophages and other immune cells. The dystrophin isoform (Dp71) found in myeloid cells remains intact in the mdx mouse[35,36], and macrophage dysregulation is thought to be primarily driven by pathological cues received directly within the local muscle microenvironment[8,12,37]. The results of the present study add a major new layer of complexity to this scenario by showing that before their entry into the pathological muscle

microenvironment, BMDM of dystrophic mice undergo extensive epigenetic remodeling which is accompanied by pronounced alterations in cellular function.

The fact that a primary muscle disease induces such effects within bone marrow myeloid cells implies a role for systemic signaling molecules such as DAMPs originating from the damaged muscles. This concept is supported by the fact that muscle extract derived not only from mdx but also from healthy WT mice, was able to induce changes consistent with trained immunity within naive WT BMDM in vitro. However, the involvement of muscle damage-derived DAMPs as the primary stimulus generating trained immunity in mdx mice remains a hypothesis, and it is possible that other factors such as cytokines or exosomes play significant roles within the in vivo environment[38,39]. Furthermore, it is interesting to note that the induction of acute necrotic muscle injury in WT mice in vivo, whether in either single or repeated bouts, did not recapitulate the changes observed in mdx BMDM. This is perhaps not surprising since the nature of the molecules released by different mechanisms of muscle damage, as well as the quantity and duration of exposure to DAMPs and other signals delivered to the bone marrow, may differ substantially for various forms of acute versus chronic muscle injury.

Several DAMPs are known to be endogenous ligands for TLR4[14]. Among candidate DAMPs that are chronically increased in the muscles and/or serum of mdx mice and DMD patients[40,41], some such as fibrinogen have been directly implicated in DMD pathogenesis[12,31,42]. The levels of such DAMPs can also vary according to disease stage[40], which may at least partially explain our observation of disease stage-specific differences in mdx BMDM characteristics. Genetic deficiency of fibrinogen led to reduced muscle pathology in mdx mice[31]. Along the same lines, we previously reported that global TLR4 deficiency in mdx mice reduces pro-inflammatory macrophages within mdx muscles and significantly mitigates disease progression including in the most severely affected muscle, the diaphragm[12]. This is consistent with the fact that low concentrations of TLR ligands induce CCR2-dependent monocyte emigration from the bone marrow[43]. In the present study, we found that the phenotypic changes observed in mdx BMDM in vitro were all largely prevented in mdx mice lacking TLR4. Although this could be partially related to less severe muscle pathology in mdxTLR4$^{-/-}$ mice, our data indicate that the ability to induce trained immunity is also severely impaired in non-dystrophic BMDM lacking TLR4. In addition, adoptive transfer studies in vivo showed that the mdx BMDM recruited to injured muscles were significantly reduced in number and associated with a less inflammatory profile when these BMDM lacked TLR4.

In our study, mdx BMDM demonstrated an undifferentiated mixed M1/M2 phenotype in vitro, whereas mdx BMDM that were adoptively transferred into an inflammatory muscle injury milieu (such as found in early stages of the disease), exhibited pro-inflammatory M1-like skewing. The alterations in cellular metabolism observed over time in mdx BMDM, suggesting a greater reliance on oxidative phosphorylation at later stages of the disease, are consistent with the macrophage phenotype evolution from M1- to M2-like previously described within mdx muscles[7,44]. We speculate that the epigenetic and metabolic reprogramming found in mdx BMDM serves to place the cells in a more conducive state for assuming different possible macrophage phenotypes after entering the muscle. According to this model (see Fig. 8), the functional modifications taking place centrally in the bone marrow favor more efficient adoption of either end (or some combination thereof) of the M1/M2 polarization spectrum. The ultimate phenotypic outcome peripherally in the dystrophic muscle tissue is determined by additional

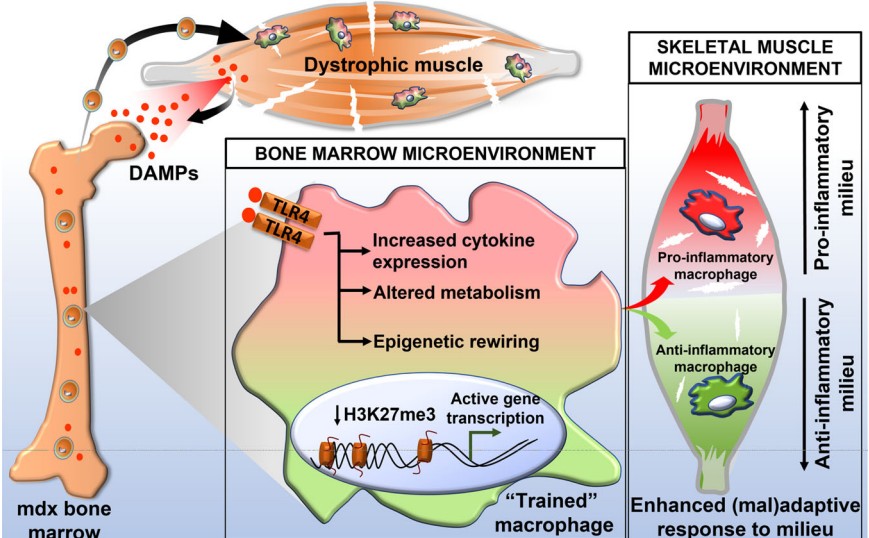

**Fig. 8 Model for cross-talk between dystrophic skeletal muscle and bone marrow leading to a reprogrammed macrophage phenotype.** According to the model, chronic systemic exposure to damage-associated molecular patterns (DAMPs) acting as ligands for innate immune receptors such as TLR4, induces the hallmark features of trained immunity (cytokine hyperresponsiveness, metabolic alterations, epigenetic rewiring) in macrophage precursor cells residing in the bone marrow. A more open chromatin state for both pro-inflammatory and anti-inflammatory genes enhances the ability of future macrophages to adopt different phenotypes once the cells home to damaged dystrophic muscle tissue. The ultimate macrophage functional phenotype and impact (adaptive or maladaptive) are dependent on the combination of signals received at the bone marrow level and within the dystrophic muscle milieu.

signals received within the local muscle microenvironment. This scenario is analogous to the epigenetically-mediated "poised" state of embryonic stem cells, where a mixture of activating and repressive marks on the same histones helps to maintain pluripotency until further environmental cues are received to drive differentiation toward a specific lineage[32].

Trained immunity is an evolutionarily conserved protective response against infection that first arose in organisms without an adaptive immune system such as plants and invertebrates[16]. In addition to epigenetic changes, we found that mdx BMDM demonstrated several other keystone features of trained immunity. A major hallmark is the ability to invoke a more intense inflammatory response upon re-challenge not only with the initial triggering stimulus, but also with other agents that differ from the initial challenge stimulus[16]. This is thought to result from the fact that a more open chromatin state induced by the epigenetic alterations permits a relatively broad and non-specific increase in transcription factor accessibility to the regulatory elements of innate immune system genes. Along these lines, mdx BMDM demonstrated increased cytokine responses to a wide range of heterologous stimuli (cytokines, fibrinogen, β-glucan) despite the reliance of these diverse stimuli upon different cell surface receptors. Analysis of classical markers for M1 and M2 macrophages suggested global activation of the cells rather than simple skewing toward M1 or M2 subtypes, which is also observed in trained immunity induced by β-glucan[21,45]. The above characteristics were found in mdx BMDM despite removal of the cells from their in vivo environment for 1 week, a "rest period" previously used to demonstrate trained innate immunity in cultured human monocytes[46,47]. Remarkably, long-lasting differences between WT and mdx BMDM were also transmissible in vivo, being maintained for at least 11 weeks after removal from the inciting muscular dystrophy environment by transplanting the mdx bone marrow into healthy WT mice.

Epigenetic changes representing a form of imprinted cellular memory or education have increasingly been implicated in the pathogenesis of inflammatory diseases[16,26]. In some cases the maintenance of inflammation at an abnormally high basal level is

sustained for a prolonged period after removal of the activating stimulus, which has been referred to as priming[17]. In other instances, removal of the inciting stimulus allows innate immune system activity to largely return to its original low basal state, but upon reintroduction of either the same or unrelated immunological stimuli there is a more robust transcriptional response. The latter situation is characteristic of trained immunity and was found to be the case in mdx BMDM. The most commonly reported epigenetic changes linked to trained immunity in prior studies have involved histone modifications such as H3K4 methylation and H3K27 acetylation[48]. In addition, it was recently shown that TLR4 signaling triggered by transient LPS exposure can induce persistent alterations of myeloid enhancer accessibility within hematopoietic stem cells, accompanied by improved innate immunity against infection[29].

In the present study, we show that mdx BMDM exhibit a complex combination of histone mark modifications that can simultaneously favor or impede the open chromatin state. These findings are similar to a previous report also showing a simultaneous decrease in repressive and activating histone marks in trained immunity[49], which could serve to maintain gene transcription in a poised state. It is also important to note that epigenetic modifications in mdx BMDM undoubtedly extend well beyond the changes found in our study. However, at least with respect to H3K27, it appears that increased chromatin accessibility in mdx BMDM is primarily mediated through reduced methylation rather than increased acetylation. Furthermore, the changes in H3K27me3 were TLR4-dependent and involved genes associated with dystrophic muscle inflammation and fibrosis.

Current evidence suggests that trained immunity can occur in humans[18–20], where it may be an important driver of pathological inflammation in non-infectious diseases such as atherosclerosis[23], Alzheimer's disease[22] and chronic allergy[24,25]. To our knowledge the present study provides the first evidence for trained immunity in DMD or any other form of skeletal muscle disease. These findings suggest the possibility of a new paradigm for DMD muscle inflammation in which monocyte-derived macrophages from the bone marrow undergo a process of extensive cellular

reprogramming before being recruited to the diseased muscles. This altered myeloid cell phenotype is regulated by TLR4 and displays the features of: (1) increased constitutive and DAMP-stimulated cytokine production, (2) changes in cellular metabolism, and (3) epigenetic remodeling. While these results are consistent with altered innate immune memory in murine DMD, whether these phenomena also occur in human DMD or other types of chronic muscle disease remains to be determined. The mdx model of DMD exhibits a more intense phase of early necrosis than human patients. In addition, the specific stimuli involved in the reprogramming of these cells are yet to be elucidated. Future studies should be directed at determining whether either preventing or reversing these processes is feasible and able to favorably modify the course of the disease. It will also be of interest to ascertain whether trained immunity, through amplification of the host immune response, represents an impediment to gene therapy and other dystrophin restoration strategies.

## Methods

**Experimental animals.** All procedures were approved by the Animal Care and Use Committee of Research Institute of the McGill University Health Centre (RI-MUHC, protocol #3480), according to the guidelines issued by the Canadian Council on Animal Care. Wild-type (WT) mice: CD45.2 (stock #000664) and CD45.1 (stock #002014) alleles, dystrophic $mdx^{4cv}$ mice (mdx, stock #002378) and TLR4$^{-/-}$ mice (stock #029015) were on the C57BL/6 J background, with all breeding pairs originally purchased from The Jackson Laboratories (Bar Harbor, ME). The mdx mice lacking TLR4 (mdxTLR4$^{-/-}$) were generated as described previously[12]. All experiments were performed on male mice. Both experimental and control groups were bred separately and co-housed under pathogen-free conditions at the animal facility of RI-MUHC and kept under a 12-h light/12-h dark cycle at a temperature $21 \pm 1$ °C and relative humidity of 40–60%. Euthanasia was conducted under anesthesia with isoflurane followed by cervical dislocation.

**Culture of bone marrow-derived macrophages (BMDM).** L929 mouse fibroblast cells were purchased from the American Type Culture Collection and grown at 37 °C with 5% CO$_2$ for 7 days in Dulbecco's Modified Eagle Medium (DMEM) supplemented with 10% fetal bovine serum (FBS); supernatant containing macrophage colony-stimulating factor (M-CSF) was then collected and stored at −80 °C for later use. For BMDM culture, tibia and femur bone were harvested bilaterally and bone marrow was flushed out using a syringe filled with Roswell Park Memorial Institute (RPMI) (Wisent) medium containing 10% FBS. Bone marrow cells were plated in RPMI media supplemented with 2 mM L-glutamine, 10% FBS, 1% non-essential amino acids, 1% essential amino acids, 0.14% 5 N NaOH, 2% HEPES, 1 mM sodium pyruvate, 100 U/ml penicillin, 100 mg/ml streptomycin (all from Wisent) and 30% L929 supernatant for 7 days in order to obtain differentiated BMDM. For stimulation, BMDM obtained at the end of 7 days were treated with LPS (100 ng/ml) + IFN-γ (20 ng/ml) (Invitrogen, USA), IL-4 (20 ng/ml) (Invitrogen, USA), fibrinogen (1 mg/ml) (Sigma, USA), or β-glucan (100 μg/ml, Sigma, USA) for 4 h and 24 h prior to cell harvesting.

**Real-time quantitative PCR (RT–qPCR).** Total RNA was extracted using TRIzol reagent (Invitrogen, USA) according to the manufacturer's protocol. After DNase I (Gibco, USA) treatment, 1 μg of RNA was used for cDNA generation with random primers and SuperScript II (Invitrogen, USA) reverse transcriptase. RT–qPCR was performed using 10 ng of cDNA mixed with 5 μl of 2x SYBR Green (Invitrogen, USA) and 0.5 μl of 10 μM primer mixes. RT–qPCR was carried out for 40 cycles at a melting temperature of 95 °C for 15 s and an annealing temperature of 60 °C for 1 min using a StepOne Plus Thermocycler (Applied Biosystems, USA). Mouse HPRT1 was used as an internal control. The relative quantification of gene expression was analyzed by the $2^{-\triangle\triangle Ct}$ method, and the results are expressed as n-fold difference relative to untreated WT control cells. Primer sequences are shown in the Supplementary Data 6 table. Gene expression from qPCR was used to generate heatmaps and perform unsupervised hierarchical clustering with a web-based program (Heatmapper[50], http://www.heatmapper.ca). Average linkage and Euclidean distance measurement methods were used for clustering the samples.

**Western blot.** After washing and scraping the cells, total cell lysate was prepared in radioimmunoprecipitation assay (RIPA) buffer followed by incubation on ice for 1 h. Following centrifugation at $16,000 \times g$ for 15 min, protein concentration was detected by Pierce™ BCA Protein Assay Kit (Thermo Fisher Scientific, USA). Equal amounts (20 μg) of protein lysate were loaded onto Mini-PROTEAN® Precast Gels (Bio-Rad, Canada) and then transferred onto polyvinylidene fluoride (PVDF) membrane. After blocking with 5% BSA in Tris buffered saline (TBS) containing 0.1% Tween-20, the membranes were incubated overnight at 4 °C with the following primary antibodies (1:1000 dilution; from Cell Signalling, USA, unless

stated otherwise): p-STAT1 (#9167), total STAT1 (#9172), p-STAT3 (#9131), total STAT3 (#9139), p-STAT6 (#9361), total STAT6 (#9362), arginase1 (#93668), and iNOS (1:1000 dilution, #610329, BD Biosciences, USA). After washing with TBS containing 0.1% Tween-20, blots were incubated with horseradish peroxidase (HRP)-conjugated anti-mouse (1:5000 dilution, #W4021) and anti–rabbit (1:2500 dilution, #W4011) secondary antibodies (Promega, Canada). Blots were developed using the enhanced chemiluminescence (ECL) system (ECL Plus; Thermo Scientific, USA). As a loading control, membranes were incubated overnight with actin antibody (1:10,000 dilution, #A2228; Sigma, USA). The original western blot images are provided in the corresponding Source Data file sections. The intensity of the bands was quantified with Image Lab software (Bio-Rad, USA), and the densitometry analysis is expressed as n-fold-change relative to WT control after actin normalization.

**Cellular bioenergetics (Seahorse) analyses.** Measurements of mitochondrial oxygen consumption rate (OCR) were obtained using the XF96 Extracellular Analyzer (Seahorse Bioscience). BMDM at a density of $1 \times 10^5$ in 100 μl of DMEM media were seeded into XF96 cell culture microplates 24 h before the assay. For the assay, XF assay media supplemented with 10 mM glucose and 1 mM sodium pyruvate at pH 7.4 was used. Following the basal respiration measurement, the OCR were analyzed after three measurements were taken pre- and post-addition of 1.5 μM oligomycin, 1 μM carbonilcyanide p-triflouromethoxyphenylhydrazone (FCCP), 2 μM rotenone, and 1 μM antimycin (all from Sigma, USA). Background from cell-free wells was subtracted, and basal and maximal OCR values were calculated using Seahorse Wave software (Seahorse Bioscience). For normalization, adherent cells were washed twice with phosphate-buffered saline (PBS) and fixed with 1% paraformaldehyde (PFA) for 20 min at 4 °C and stained with 0.05% crystal violet for 30 min. After washing away the excess crystal violet with PBS, the stained cells were solubilized with 1% sodium dodecyl sulfate (SDS) followed by optical density (OD) measurements at 595 nm using a Tecan plate reader. Data are expressed as OCR normalized to the OD value from each individual well.

**Lactate release assay and ELISA.** BMDM were cultured in 6-well plates in phenol-red-free RPMI media for 48 h for lactate assays. Supernatants were collected and centrifuged at $16,000 \times g$ for 15 min to remove cellular debris before storing at −80 °C for later use. Supernatant lactate levels were measured with a commercial kit following the manufacturer's instructions (Lactate Assay Kit, Eton Bioscience Inc, USA). Lactate concentrations were normalized to the cellular content of each well as reflected by total protein levels after removal of supernatant and washing of the cells with PBS. For ELISA, BMDM were cultured in RPMI media containing fibrinogen (1 mg/ml) for 24 h before collecting the supernatant. Levels of TNF and IL6 in supernatant were measured by ELISA kit according to the manufacturer's instructions (R&D Systems, USA).

**BMDM training with muscle extract and serum.** Muscle extract and serum were used as primary training agents for WT BMDM. Muscle extracts were prepared fresh from WT and mdx muscle by homogenizing the muscle in PBS. Following centrifugation at $13,000 \times g$ for 15 min, extracts were collected and passed through 0.2 μm filter before measuring the protein concentration. Similarly, serum from 4 to 6 weeks old WT and mdx mice was collected and filter-sterilized prior to use. At day 3 of cell culture, the cells were incubated in RPMI media containing muscle extract (1 mg/ml) or 5% serum or PBS for 24 h. Cells were washed twice with RPMI media before allowing them to rest for 5 days undisturbed except the addition of fresh RPMI media. BMDM were then secondarily restimulated with fibrinogen (1 mg/ml) for 8 h or β-glucan (100 μg/ml) for 4 h, followed by collection of the cells for RNA extraction.

**Generation of bone marrow chimeric mice.** C57BL/6 CD45.1 (Jackson Laboratory, USA) recipient mice (6 weeks of age) were lethally irradiated with total 12 Gy (delivered with two doses of 6 Gy at 4 h apart) (X-RAD smart, Precision X-ray, North Branford, CT). At 24 h after the second dose of radiation, $4 \times 10^6$ bone marrow cells isolated from either WT or mdx (donor, CD45.2 mice) were injected intravenously to recipient mice. Mice were given antibiotic treatment (50 mg/ml enrofloxacin, Bayer, USA) in drinking water for three days before and 2 weeks after irradiation. Bone marrow chimerism was determined by flow cytometry at 11 weeks after transplantation.

**Bone marrow adoptive transfer and acute muscle injury.** For adoptive transfer experiments, bone marrow of CD45.2 allele donor mice (WT, mdx or mdxTLR4$^{-/-}$) was isolated and $6 \times 10^7$ cells in PBS were injected intravenously into CD45.1 allele host recipient mice immediately prior to induction of muscle injury. To induce acute skeletal muscle injury, both hindlimbs (tibialis anterior (TA) muscle) were injected with 25 μl of 10 μM cardiotoxin as previously described[51]. Four days later the recipient mice were euthanized and injured muscles were processed to separately identify the host and donor-derived macrophages according to their CD45 allelic status by flow cytometry as described below.

**Flow cytometry**. Single cell suspensions were obtained by incubating minced muscles in 0.2% collagenase (Roche) at 37 °C for 45 min followed by filtering with a 70-μm cell strainer. Bone marrow was flushed out using a syringe filled with PBS and passed through 70-μm cell strainer. For flow cytometry, $10^6$ cells were resuspended in PBS containing 0.5% BSA (FACS buffer). To distinguish between live and dead cells, cells were stained with AF700 viability dye (Thermo Fisher Scientific, USA) for 30 min at 4 °C. The cells were then blocked with Fc blocking solution (anti-CD16/CD32, BD Biosciences) for 10 min followed by fluorescent-labeled antibody staining for 30 min at 4 °C.

Antibodies for flow cytometry were used at 1:100 dilution (BD Biosciences, USA) unless stated otherwise. For cell surface markers, the cells were stained at 4 °C in the dark with primary antibodies: PE-labeled anti-CD45.1 (clone A20), V500 labeled anti-CD45.2 (clone 104), BUV395 labeled anti-CD45.2 (clone 104), BUV737 labeled anti-CD11c (1:50 dilution, clone HL3), APC-Cy7 labeled anti-CD11b (clone M1/70, BioLegend), PE-Cy7 labeled anti-F4/80 (clone BM8, BioLegend), biotin conjugated: CD5 (clone 53-7.3, BioLegend), anti-Ly6G/C (clone RB6-8C5, BioLegend), anti-CD11b (clone M1/70), anti-CD4 (clone RM4-5), anti-CD8a (clone 53-6.7), anti-CD45R (clone RA3-6B2), Streptavidin conjugated APC-Cy7, BV785 labeled anti-CD127 (clone A7R34), PE-Cy7 labeled anti-Sca-1 (clone D7, BioLegend), APC labeled anti-c-Kit (clone 2B8), FITC labeled anti-CD34 – FITC (clone RAM34), PerCp-Cy5.5 labeled anti-CD16/32 (clone 93), PerCP-Cy5.5 labeled anti-CD3 (clone 145-2C11), PE-Cy7 labeled anti-CD4 (clone 53-6.7, BioLegend), APC labeled CD8 (clone 53-6.7), BUV395 labeled anti-CD19 (clone 1D3), FITC labeled anti-Ly6G (Clone 1A8, BioLegend), APC labeled anti-Ly6C (Clone HK1.4, BioLegend), and PE-CF594 labeled anti Siglec-F (Clone E50-2440).

After staining, cells were washed three times with FACS buffer followed by fixation with 4% PFA. For the staining of intracellular markers, cells were permeabilized with 1X eBioscience™ permeabilization buffer (Invitrogen, USA) and stained with antibodies in the permeabilization buffer at 4 °C in the dark: FITC labeled anti-iNOS (clone 6/ iNOS/NOS Type II), BV421 labeled anti-TGFβ (clone TW7-16B4). After washing three times cells were resuspended in 300 μl of FACS buffer and 200,000 events were acquired on a BD LSRFortessa X-20 machine using BD FACSDiva™ Software. Finally, the data were analyzed using FlowJo software (Treestar Inc., Ashland, USA). Gating strategies (Supplementary Fig. 12) were based on the FMO (Fluorescence Minus One), which includes all antibodies but the one for which gating is intended.

**Chromatin immunoprecipitation (ChIP)-qPCR**. Native chip assay was performed as described previously[52]. Cultured BMDM were washed twice with cold PBS and cells were non-enzymatically detached from the plate using Corning™ CellStripper Dissociation Reagent (Corning, USA). Cells were washed three times with cold PBS and $5 \times 10^6$ cells were then resuspended in douncing buffer (10 mM Tris-HCl, pH 7.5, 4 mM MgCl2, 1 mM CaCl$_2$ and 1x protease inhibitor cocktail (PIC)) and homogenized through a syringe. Chromatin was digested in Micrococcal nuclease (New England Biolabs, Canada) at 37 °C for 7 min, and the reaction was inactivated by 0.5 M EDTA. Chromatin was resuspended in hypotonic buffer (0.2 mM EDTA, pH 8.0, 0.1 mM benzamidine, 0.1 mM phenylmethylsulfonyl fluoride, 1.5 mM dithiothreitol and 1x PIC) and incubated for 1 h on ice. After spinning down the cell debris the supernatant was recovered. Chromatin was pre-cleared with 100 μl of protein A Dynabeads (Invitrogen, USA) and immunoprecipitation was carried out by incubating with a complex of beads and H3K4me3 antibody (1:50 dilution, #ab8580; Abcam, UK) or H3K27me3 antibody (1:50 dilution, #9733; Cell Signalling, USA) or H3K27Ac antibody (1:50 dilution, #39133; Active Motif, USA,) or control IgG antibody (1:50 dilution, #3900; Cell Signalling, USA) overnight at 4 °C. The complexes were washed twice with ChIP wash buffer I (20 mM Tris-HCl, pH 8.0, 0.1% SDS, 1% Triton X-100, 2 mM EDTA and 150 mM NaCl) and once with 400 μl of ChIP wash buffer II (20 mM Tris-HCl (pH 8.0), 0.1% SDS, 1% Triton X-100, 2 mM EDTA and 500 mM NaCl). Finally, protein–DNA crosslinks were reversed in 200 μl of elution buffer (100 mM NaHCO$_3$ and 1% SDS) for 2 h at 68 °C and the isolated DNA was purified using phenol chloroform. ChIP DNA was analyzed by qPCR using primers specific to promoter regions of interest, and the data were normalized by input DNA. The primers used for ChIP-qPCR are listed in the Supplementary Data 6 table.

**Library preparation for ChIP sequencing**. For library preparation, ChIP DNA ends were first repaired at 20 °C for 30 min with repair buffer (2.9 μl of H$_2$O, 0.5 μl 1% Tween-20, 1 μl dNTP mix 10 mM (#77119, Affymetrix, USA), 5 μl 10x T4 ligase buffer (#L6030-HC-L Enzymatics, Qiagen, USA), 0.3 μl T4 DNA pol (#P7080L Enzymatics), 0.3 μl T4 PNK (#Y9040 Enzymatics) and 0.06 μl Klenow (#P7060L Enzymatics). Next, a mixture of 1 μl of Seradyn 3 EDAC SpeedBeads (#6515-2105-050250, Thermo Fisher Scientific) and 93 μl of 20% PEG8000/2.5 M NaCl (13% final) was added and incubated for 10 min. Magnetic beads were washed twice with 80% ethanol and air-dried for 10 min before elution in 15 μl ddH$_2$O. dA-Tailing was performed by incubating DNA at 37 °C for 30 min in the mixture of solution with 10.8 μl ddH2O, 0.3 μl 1% Tween-20, 3 μl Blue Buffer (#B0110L Enzymatics), 0.6 μl dATP 10 mM (#10216-018, Thermo Fisher Scientific), 0.3 μl Klenow 3-5 Exo (#P7010-LC-L Enzymatics). Furthermore, 20% PEG8000/2.5 M NaCl (13% final) was added and incubated for 10 min. Beads were eluted in 14 μl of elution buffer (Zymo Research, USA). For adapter ligation, sample was mixed with 0.5 μl of a BIOO barcode adapter (#514104 BIOO Scientific, USA) with 15 μl Rapid Ligation

Buffer (#L603-LC-L Enzymatics), 0.33 μl 1% Tween-20 and 0.5 μl T4 DNA ligase HC (#L6030-HC-L Enzymatics) and incubated for 15 min at room temperature. Beads were further cleaned with 7 μl of 20% PEG8000/2.5 M NaCl. Lastly, beads were eluted in 21 μl of elution buffer. From the eluted volume, 10 μl was further used for PCR amplification (16 cycles) with IGA primers (AATGA-TACGGCGACCACCGA) and IGB primers (CAAGCAGAAGACGGCATACGA) in 1:1 ratio.

**ChIP-sequencing data analysis**. All H3K27me3 reads for the three libraries (WT, mdx, mdxTLR4$^{-/-}$) were first mapped to the mm10 genome using Bowtie2[53] under the default parameters. There were four samples for each of the libraries (H3K27me3 replicate 1, H3K27me3 replicate 2, Input replicate 1, Input replicate 2). The reads from replicates were merged for the downstream analyses. With the mapped reads, we quantified the read intensity for both the H3K27me3 and Input sample of each library using deepTools[54]. The methylation peaks were also called from the mapped reads using Spacial Clustering for Identification of ChIP-Enriched Regions (SICER)-2[55] with the default parameters (window size = 200, gap size = 600). All peaks with FDR < 0.01 were kept as the final significant peaks. The overlapped genes were merged and finally 35,918 significant peaks for WT, mdx, and mdxTLR4$^{-/-}$ were identified.

After all the peaks and genome-wide read intensities for the three libraries were acquired, different methylation patterns for these peaks were assigned. We first determined whether there was a significant methylation difference (H3K27me3 vs. Input) associated with the peak region for each of the libraries. The significance was calculated based on the read intensity difference between the H3K27me3 and Input data using a one-sided Mann–Whitney U-test. If the H3K27me3 read intensity was significantly higher (defined as P-value < 0.05 and intensity fold-change >1.5) than its corresponding input, the peak was designated as Increased (I); otherwise, the peak was considered as Unchanged (U). Using these designations, all the obtained peaks were assigned to 8 ($2^3$) different patterns for WT, mdx, mdxTLR4$^{-/-}$ libraries included in this study. After inferring the pattern for a specific peak relative to the input, we also required that the methylation intensity fold-change of the peak in any "Increased (I)" library be statistically greater than that in any "Unchanged (U)" library (Mann–Whitney U-test P-value < 0.05). The associated genes were identified for these significant peaks if located within –5 kb to +1 kb of the gene.

Besides the above analysis strategy, the patterns for all genes across the genome were also directly evaluated as follows. If the methylation read intensity in the nearby region (–5 kb to +1 kb relative to the gene body[56]) was significantly greater in the H3K27me3 sample than its corresponding Input, this gene was designated as Increased (I); otherwise, it was considered as unchanged (U). Using this approach, defined as the Gene-based Pattern (GP) analysis, we identified 3299 genes with 4 distinct patterns (designated GP1 to GP4 = IUI, IUU, UII, and UIU, respectively) of dynamic regulation in the mdx group relative to WT (GP1: 2293 genes, GP2: 786 genes, GP3: 154 genes, GP4: 66 genes). This analysis allowed the application of biological pathway enrichment analysis (see below) for any given list of genes (e.g., genes from a particular pathway).

To assess whether a given list of genes was statistically enriched with a particular pattern, we used the binomial test as follows. First, we calculated a background probability $p_b(pt)$ for each pattern pt, which is learned as the percentage of genes (out of all genes) that follow that particular pattern pt. The enrichment P-value can be calculated using the following Eq. 1:

$$P \text{ value} (pt) = 1 - \sum_{i=0}^{K} \binom{i}{n} p_b(pt)^i \left(1 - p_b(pt)\right)^{n-i}, \tag{1}$$

where $n$ is the total number of genes in the gene list of interest, and $k$ is the number of genes that follow the pattern pt (the null hypothesis H0: more than $k$ out of $n$ genes follow the pt pattern randomly). The biological pathways enriched with the above patterns were determined using the Reactome pathway database[57], with a particular focus on those four patterns showing dynamic regulation in WT vs. mdx groups (GP1–GP4). Using the approach described above, a list of enriched Reactome pathways (sorted by the binomial test P-value) associated with each of these patterns (both peak-based and gene-based) was obtained.

In addition, Hypergeometric Optimization of Motif EnRichment (HOMER)-2 software[58] was employed to identify potential transcription factor enrichment within the peak regions exhibiting dynamic differences between the WT and mdx groups. Using an analogous approach to the Gene-based Pattern analysis outlined earlier, this was designated as a Peak-based Pattern (PP) analysis (PP1 to PP4 = IUI, IUU, UII, and UIU patterns, respectively). This information then allowed further exploration of the biological functions linked to genes in proximity (–5 kb to +1 kb of the gene) to the peaks using complementary Reactome pathway analysis.

The H3K27ac ChIP-seq data analysis followed the same general procedure as the H3K27me3 analysis described above. However, since H3K27ac primarily marks the promoters and enhancers located in the upstream region of the gene, only the upstream region (–5 kb) was targeted for the Gene-based Pattern analysis. In addition, known enhancer annotations[59] were considered when mapping peaks to nearby genes.

**Statistics and reproducibility**. Unless stated otherwise, statistical analyses were performed using GraphPad Prism Version 6.01 (San Diego, CA, USA). A $P$-value < 0.05 was considered to be statistically significant (two-tailed for all data). Comparisons between two groups were determined by unpaired $t$-test, and between more than two groups by one- or two-way ANOVA followed by a Tukey post-hoc test to adjust for multiple comparisons. Outliers identified by the software were excluded from the analysis. Error bars represent standard error of the mean (SEM) for the indicated number of observations. The "$n$" for each bar graph represents the number of independent biological replicates. For each figure, the statistical test used and number of observations are shown in the graph and respective figure legend. Exact $P$-values for each individual group comparison are provided in the Source Data file.

**Reporting summary**. Further information on experimental design is available in the Nature Research Reporting Summary linked to this paper.

## Data availability

The primary data which support the findings of this study are available in the Source Data file accompanying the paper. ChIP-seq data have been deposited in the Gene Expression Omnibus database under accession number GSE190307. We also provide a data server to enable interactive queries of the H3K27me3 status for any gene list (https://junding.lab.mcgill.ca/research/basil/EnrichmentServer/). Similarly, a webserver for interactive exploration of H3K27ac status is also provided (https://junding.lab.mcgill.ca/research/basil/H3K27ac/EnrichmentServer/). Source data are provided with this paper.

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

## Acknowledgements

This work was supported by the Canadian Institutes of Health Research (FRN 148744 to B.J.P.) and the JT Costello Memorial Fund of the Meakins-Christie Laboratories. We thank Dr. Carol CL Chen from Department of Human Genetics, McGill University for kind assistance with ChIP-qPCR experiments.

## Author contributions

B.J.P. conceived and directed the project. S.B., Q.L., F.L., E.G., G.J.F., J.D., and E.K. developed methods and/or performed experiments. S.B., Q.L., O.L., G.J.F., E.K., and J.D. performed data analysis. S.B., M.D., J.D., and B.J.P. developed experimental concepts. S.B., J.D., and B.J.P. wrote the manuscript.

## Competing interests

The authors declare no competing interests.
