## [Peer Review File · Nature Communications]

TLR4 is a regulator of trained immunity in a murine model of
Duchenne muscular dystrophyREVIEWER COMMENTS

Reviewer #1 (Remarks to the Author):

In this manuscript, the Petrof lab reported trained immunity in DMD pathogenesis. The lab has an excellent track-record in dissecting immunological aspect of DMD pathogenesis. The study described here represents an important advance and provides critical new insight in understanding DMD inflammation. The manuscript is well-written and data are of high quality.

Major

1. The acute necrotic phase is unique to the mouse model. This phase does not exist in human patients. It appears that the release of DAMPS from massive necrotic damage during the acute necrotic phase plays an important role in the formation of the trained immunity in mdx BMDM. In the absence of data from human DMD patients or large animal models (such as GRMD model), I'd suggest authors to tone down the conclusion.
2. Considering inflammation is a generic feature in many types of muscular dystrophy, I'm wondering whether the authors have confirmed their findings in a mouse model of different type of muscular dystrophy (such as LGMD).
3. Results from Figure 4 suggest that "Muscle tissue damage per se without the associated inflammation of dystrophic muscles was sufficient to induce" phenotypic reprogramming of mdx BMDM. This is contradictory to the finding that repeated cardiotoxin injection in WT muscle failed to reproduce the metabolic changes in mdx BMDM. The authors speculated that "differences in the nature, timing or quantity of DAMPs and other signals delivered to the bone marrow are likely responsible for the divergent bone marrow myeloid cell responses to acute versus chronic muscle injury". I'm wondering whether authors imply that their findings are unique to the DMD mouse model.
4. I'm curious if therapeutic intervention (such as systemic micro-dystrophin gene therapy) in post-necrotic phase (~10-week-old) or fibrotic phase (1-year-old) would impact trained immunity in mdx BMDM.
5. Figure 4d-h. I'm wondering if the authors have tested wt donors that have undergone cardiotoxin injection. I'm also curious whether transfer of mdx BMDM resulted in any muscle pathology in the recipient mouse.
6. Figure 5b. In the panel of necrotic phase, mice of the same genotypes were grouped together. This makes it easy to see the difference. However, the the panels of pre-necrotic and fibrotic phases, mice are not organized in the same way. This makes it difficult to compare the changes at three different phases in three groups of mice (wt, mdx, mdx/TLR4KO).
7. TLR4 KO largely prevented trained immunity in mdx BMDM. I'm curious whether muscle disease in mdx/TLR4 KO mice is ameliorated. Some evidence in this regard will help determine whether modulation of trained immunity represents a viable therapeutic approach.

Minor

1. In the introduction sentence, please cite a reference on "Duchenne muscular dystrophy" (e.g. PMID: 33602943) . This will be useful for readers who are not familiar with the disease.
2. Please cite a reference to support the sentence "Dystrophin deficiency is necessary but not sufficient on its own to fully account for the relentless course of muscle necrosis and fibrosis in DMD".
3. Line 58 "However, the cellular and mechanisms underlying this". please remove "and".
4. Figure 6, panel b. please label which is which. I assume the first column is wt, the second mdx and third mdx/TLR4 KO.

Reviewer #2 (Remarks to the Author):

In this interesting research paper, the authors present the results of a series of in vitro and in vivo mouse studies that together support the hypothesis that trained immunity at the level of bone marrow myeloid progenitors contributes to the progressive inflammatory muscle damage in Duchenne muscular dystrophy (DMD).

This is a very interesting and timely hypothesis which could potentially unveil novel potential pharmacological targets to prevent progression of this devastating disease. I have several major and minor comments.

MAJOR COMMENTS:

1. Dystrophin is also expressed in the myeloid lineage. Hence, it cannot be excluded that the inflammatory phenotype of the monocytes and progenitor cells results from a direct effect of this genetic deficiency in addition to the effect of DAMPS from necrotic muscle on these cells. Even the fact that the authors show that early in the course of the disease, there is no activation of BMDMs this doesn't exclude that the lack of dystrophin in these cells contribute to the temporal changes in activation status. There are several experiments that could provide these data, e.g. BMT from WT mouse into mdx mice; showing that also the WT progenitor will show trained immunity.
2. To increase the impact of their findings, it would further strengthen the data if the authors could demonstrate markers of trained immunity also in human circulating monocytes from patients with DMD (e.g. increased cytokine production capacity, typical histone modifications).
3. The main read out for 'trained immunity' is the enhanced cytokine production capacity. The authors only measure mRNA expression throughout the manuscript. Please also provide data on extracellular cytokine protein concentrations.

MINOR COMMENTS:

1. Lines 60-68 describe the M1-M2 paradigm of macrophage polarization. First, I would recommend to describe that this is a highly simplified model and that there are multiple phenotypic states dependent on the specific micro-environment (Xue, Immunity 2014). Also, the authors use the stimuli that are classically used for M1/M2 differentiation to restimulate the cells for 24 hours. However, this is not differentiation, but restimulation of trained cells. Why do the authors use these stimuli?
2. Figure 2: please add what injections were given in the OCR-time graphs.
3. Fig 2: please also provide the lactate data for the pre-necrotic phase.
4. Lines 140-142: the authors conclude that the major difference is acute versus chronic muscular injury. Another difference could be that the mechanism of muscle cell injury (and hence the DAMPS released) is completely different between the two models.
5. Figure 3: the authors stimulate BMDMs with LPS/IFN or IL4 for 4 and 24 hours and describe M1- and M2 polarization in the result section. However, this is not a model for M1/M2 differentiation, but just restimulation of (trained/untrained) cells; 4 hours is too short for M1/M2 differentiation.
6. Page 8, why did the authors restimulate with beta-glucan? In the context of trained immunity, this is a classical stimulus used to induce training, but this is never used as stimulus for restimulation. Please explain.
7. Page 9: BMDMs are exposed to crushed skeletal muscle extracts. If the authors hypothesize that DAMPS from these muscles are released into the blood stream to affect bone marrow progenitors, it would also be possible to induce training by exposing BMDMs to serum from the mdx mice versus WT mice. This could also be done with serum from patients with DMD.
8. Why did the authors restimulate with fibrinogen? This is an endogenous compound present in high concentrations in the circulation. Does this trigger cytokine release in immune cells?
9. On page 10, the authors describe the results of the mdxTLR4^{-/-} mice and show less inflammatory reprogramming of the BMDMs. Since TLR4 deficiency is not restricted to the myeloid lineage, I can imagine that TLR4^{-/-} also leads to a lower degree of muscular necrosis/inflammation/DAMP release. The authors should show that the stimulus (DAMPS) that allegedly reprograms the progenitor cells is unaffected in these mice.
10. Page 11: please explain why you focused on K27me3, compared to other markers classically linked to trained immunity (K4me3, K4me, K27Ac).

Reviewer #3 (Remarks to the Author):

In this manuscript, the authors describe the involvement of trained immunity in the pathogenesis of Duchenne muscular dystrophy (DMD). Firstly, the authors utilized bone marrow derived macrophages (BMDMs) and conducted a series of real time PCR assays. They found that, in the mdx mice, which is a model of DMD, authentic features of trained immunity was observed. In the mdx mice, trained immunity was hypersensitive compared to the wild type mice. Those responses were invoked by muscle extract, suggesting its important role in etiology of DMD. They also examined and found that all the molecular responses and phenotypic appearance were TLR4 dependent. Importantly, the authors conducted an in vivo experiment and found that the hypersensitive phenotype was transmissible by bone marrow transplantation to healthy mice. The authors also describe that the training of this innate immune response should be mediated by the epigenetic modifications of particular genomic regions in BMDMs. Overall, I fully agree that this paper precisely aiming at the molecular features underlying this difficult disease. In fact, the training mechanisms of the immune system in DMD remains almost totally unknown and its better understandings may open a new research field for drug development or designing other therapeutic strategies. However, after reading through this manuscript, I still have remaining concern what is the essence of the molecular memory of this phenotype of the mdx mice. The presented data is mostly from the rather straightforward gene expression analyses and the complex or heterogeneous nature of its association with epigenome elements are still remaining uncharacterized.

Major points:

1. It is essential to further elucidate the epigenomic features of BMDMs. Indeed, Figure 6e clearly represents the “changes” in epigenomic features in the wt. mdx and mdx/TLR4(-/-) mice. However, this data also indicates that there should be certain cellular heterogeneity within the population. Considering there are only two copies of genomic DNA per cell, the observed changes should represent the changes in the cellular populations having the corresponding epigenomic patterns. This heterogeneity may be further associated with the separation of BMDM potentials in pro-inflammatory or anti-inflammatory milieus. To further elucidate these features, I consider the single cell analysis should be essential. Particularly, the simultaneous measurement of the gene expressions and the open chromatin structures of a give cell has become rapidly easy and popular.
2. Also, the status changes and the timing of the establishment for the immune cell training should be further clarified. For example, in the mdx mice, would not pre-nectoric BMDM show any responses to the stimulation?
3. Phenotypic appearances of the mice, such as the degree of the muscle destruction, should be more quantitatively analyzed. From a clinical viewpoint, it is essential to learn to what extent trained immunity is involved in the pathology of DMD. Such an information is particularly important, when the development of a new drugs or other therapeutic strategy should be considered in this direction.

Minor point:

4. It is essential to further analyze how TLR4 should transduce the signal to the downstream molecules. Some pathway analyses are described but no validation is conducted. Also, it should be validated how the mRNA changes may lead to the protein level inductions for the examined pro-inflammatory, anti-inflammatory and other molecules.
5. It is remaining elusive, at least to me, how the innate immune system may have acquired a specificity, if any, to the muscle extract, assuming there should be other antigens, particularly when we think about the case in human patients.
6. Extensive analyses is needed to further inspect the changes of H3K4me3 patterns. Particularly, I wonder whether the H3K4me3 pattern did not change around the transcriptionally activated genes or the activated genes were only minor. If the former was the case, what is the suspected controlling factor?

7. In fact, trained immunity has been indicated to be involved in other chronic diseases and cancers (Netea et al Nature Reviews Immunology 2020) in addition to DMD. Discussion should be enriched from such a viewpoint.

8. p28: Data analysis of CHIP seq should be conducted in a more quantitative manner.

9. Title is somewhat ambiguous to represent the contents of the manuscript.

10. If any of the experimental procedures was hinted by a previous paper, please refer to it (for example, de Laval et al, Cell Stem Cell, 2020?).

Reviewer #1 (Remarks to the Author):

In this manuscript, the Petrof lab reported trained immunity in DMD pathogenesis. The lab has an excellent track-record in dissecting immunological aspect of DMD pathogenesis. The study described here represents an important advance and provides critical new insight in understanding DMD inflammation. The manuscript is well-written and data are of high quality.

Response: *We thank the reviewer for these encouraging comments.*

Major

1. The acute necrotic phase is unique to the mouse model. This phase does not exist in human patients. It appears that the release of DAMPS from massive necrotic damage during the acute necrotic phase plays an important role in the formation of the trained immunity in mdx BMDM. In the absence of data from human DMD patients or large animal models (such as GRMD model), I'd suggest authors to tone down the conclusion.

Response: *We agree with the reviewer that our findings in the mouse model cannot be automatically assumed to be present in human DMD patients. As suggested by the reviewer, we have softened our conclusion in the final paragraph of the revised Discussion to reflect this fact.*

2. Considering inflammation is a generic feature in many types of muscular dystrophy, I'm wondering whether the authors have confirmed their findings in a mouse model of different type of muscular dystrophy (such as LGMD).

Response: *At this time we have not explored other models of muscular dystrophy, but this is certainly an area of interest for the future. Depending upon the timing, chronicity and magnitude of muscle necrosis, one could expect similar findings in other forms of chronic muscle disease. In reference to points 1 and 2 raised by the reviewer, we have expanded upon these concepts to our revised Discussion and included the statement "whether these phenomena also occur in human DMD or other forms of muscular dystrophy remains to be determined".*

3. Results from Figure 4 suggest that "Muscle tissue damage per se without the associated inflammation of dystrophic muscles was sufficient to induce" phenotypic reprogramming of mdx BMDM. This is contradictory to the finding that repeated cardiotoxin injection in WT muscle failed to reproduce the metabolic changes in mdx BMDM. The authors speculated that "differences in the nature,

timing or quantity of DAMPs and other signals delivered to the bone marrow are likely responsible for the divergent bone marrow myeloid cell responses to acute versus chronic muscle injury". I'm wondering whether authors imply that their findings are unique to the DMD mouse model.

Response: These are excellent points. We do not believe our findings are necessarily unique to dystrophin deficiency and were initially surprised that repeated cardiotoxin injection did not induce the trained phenotype in bone marrow-derived macrophages of wild-type (WT) mice. However, in retrospect this finding is perhaps not unexpected since we know that the muscle regeneration response to acute cardiotoxin injury is very successful with little induction of fibrosis even in mdx mice^{1, 2}, whereas chronic repetitive microtrauma in mdx muscles significantly exacerbates fibrosis³. In addition, we speculate that the absolute level of DAMP exposure (cumulative dose) is greater in mdx mice since the majority of skeletal muscles in the body contain necrotic/damaged fibers over a more sustained period of time, whereas muscle damage in the acute cardiotoxin model is transient and limited to the injected muscles.

To directly address these issues of dosage and timing, we have performed additional experiments using the in vitro model of muscle extract stimulation of WT bone marrow-derived macrophages (BMDM) as shown in Figure 4a of the original manuscript. In the revised manuscript these new experiments are presented in Supplemental Figure 4c-d, showing that induction of the trained immunophenotype in WT BMDM is dependent upon both the muscle extract concentration and the duration of exposure. This finding is consistent with prior studies of trained immunity induced by other types of primary training stimuli⁴. We now discuss in greater detail the issues raised by the reviewer in the revised manuscript (paragraph 2 of Discussion section).

4. I'm curious if therapeutic intervention (such as systemic micro-dystrophin gene therapy) in post-necrotic phase (~10-week-old) or fibrotic phase (1-year-old) would impact trained immunity in mdx BMDM.

Response: This is a very interesting question. One would predict that dystrophin restoration, if sufficiently widespread, could reduce necrosis and thus decrease the level of bone marrow stimulation by DAMPs. However, as implied by the reviewer's question, the extent to which trained immunity is reversible might differ depending upon the age of the animal. In addition, trained immunity could potentially represent a significant impediment to dystrophin gene therapy by amplifying adverse immune responses against either gene therapy vectors or the dystrophin protein itself. These are all clinically relevant questions which will require extensive experimentation to properly address and are thus beyond the scope of the current study. In the revised manuscript we now mention these

issues as important areas for future investigation (last paragraph of Discussion).

5. Figure 4d-h. I'm wondering if the authors have tested wt donors that have undergone cardiotoxin injection. I'm also curious whether transfer of mdx BMDM resulted in any muscle pathology in the recipient mouse.

Response: We interpret the reviewer's first question as being directed at whether acute cardiotoxin injury may have induced trained immunity. In the original manuscript we argued against this possibility, since we showed that in WT mice that have undergone prior bouts of acute cardiotoxin injury there were no changes in either: 1) the basal levels of M1/M2 gene expression (Figure 2e) or 2) the metabolic oxygen consumption profile (Figure 2f) of their BMDM. However, in response to the reviewer's comments we have done further experiments. To firmly establish whether acute cardiotoxin injury in WT mice induces the trained phenotype, BMDM were harvested from cardiotoxin-injured WT mice and secondarily stimulated to look for the amplified responses which are a defining feature of trained immunity. These experiments clearly demonstrate that previous cardiotoxin injection does not amplify the inflammatory gene responses to secondary stimulation of BMDM. Therefore, these new data (now shown in Supplemental Figure 3d-e) essentially rule out induction of trained immunity by the acute cardiotoxin muscle injury model.

To address the reviewer's second question of whether transfer of mdx bone marrow induces muscle pathology in WT recipient mice, we have now evaluated the macrophages present in the skeletal muscles of such chimeric mice by flow cytometry. These studies did not reveal any differences in skeletal muscle macrophage numbers between WT recipient mice transplanted with either WT or mdx bone marrow. Therefore, the transplanted mdx BMDM did not induce inflammatory pathology in the host WT skeletal muscle. These new data are presented in Supplemental Figure 4f of the revised manuscript.

6. Figure 5b. In the panel of necrotic phase, mice of the same genotypes were grouped together. This makes it easy to see the difference. However, the the panels of pre-necrotic and fibrotic phases, mice are not organized in the same way. This makes it difficult to compare the changes at three different phases in three groups of mice (wt, mdx, mdx/TLR4KO).

Response: The gene expression data in Figure 5b were analyzed using the well established and unbiased method of unsupervised hierarchical clustering⁵. This means that the order of the samples is not determined by the user but by an algorithm which builds a dendrogram based on similarity. If samples cluster and

are therefore closer to each other, it means they are more similar. This is visually represented by the dendrogram and associated heatmaps shown in the figure 5b. The height of the vertical line to the branch point on the dendrogram represents the dissimilarity between samples. If a group of samples has very little difference in height between the branch points on the dendrogram, then they are highly similar. Therefore, samples with the most similar characteristics will be placed closer to each other on the heatmap. Accordingly, the distinct genotype groupings observed in the necrotic phase (with mdx and mdxTLR4 samples clustered at opposite ends of the heatmap and all WT samples in the middle) is a reflection of the clear biological differences between these genotypes during this phase of the disease. Conversely, there are no distinct groupings during the pre-necrotic phase, which reflects a lack of evident biological differences between genotypes during this phase of the disease. Results for the fibrotic phase are intermediate. In the revised manuscript, we have attempted to better clarify the above points in the Results section under the heading "**Phenotypic reprogramming of mdx BMDM is TLR4-dependent**".

7. TLR4 KO largely prevented trained immunity in mdx BMDM. I'm curious whether muscle disease in mdx/TLR4 KO mice is ameliorated. Some evidence in this regard will help determine whether modulation of trained immunity represents a viable therapeutic approach.

Response: In a previous publication, we showed that skeletal muscle pathology is less severe in mdx mice lacking TLR4⁶. In the revised manuscript we have attempted to place this prior report and the new findings of this paper into greater context with respect to their potential therapeutic implications (paragraph 3 of Discussion section).

Minor

1. In the introduction sentence, please cite a reference on "Duchenne muscular dystrophy" (e.g. PMID: 33602943) . This will be useful for readers who are not familiar with the disease.

Response: This has been done, thank you for the suggestion.

2. Please cite a reference to support the sentence "Dystrophin deficiency is necessary but not sufficient on its own to fully account for the relentless course of

muscle necrosis and fibrosis in DMD".

Response: This statement alludes to the fact that certain skeletal muscles are relatively spared from pathology in DMD. However, because this point is not particularly relevant to our main hypothesis and may simply create confusion on the part of the general readership, we have elected to delete this sentence from the revised manuscript.

3. Line 58 "However, the cellular and mechanisms underlying this". please remove "and".

Response: Done, thank you.

4. Figure 6, panel b. please label which is which. I assume the first column is wt, the second mdx and third mdx/TLR4 KO.

Response: Thank you for detecting this omission, we have made the correction.

Reviewer #2 (Remarks to the Author):

In this interesting research paper, the authors present the results of a series of in vitro and in vivo mouse studies that together support the hypothesis that trained immunity at the level of bone marrow myeloid progenitors contributes to the progressive inflammatory muscle damage in Duchenne muscular dystrophy (DMD).

This is a very interesting and timely hypothesis which could potentially unveil novel potential pharmacological targets to prevent progression of this devastating disease. I have several major and minor comments.

Response: We thank the reviewer for these encouraging comments.

MAJOR COMMENTS:

1. Dystrophin is also expressed in the myeloid lineage. Hence, it cannot be excluded that the inflammatory phenotype of the monocytes and progenitor cells results from a direct effect of this genetic deficiency in addition to the effect of DAMPS from necrotic muscle on these cells. Even the fact that the authors show that early in the course of the disease, there is no activation of BMDMs this doesn't exclude that the lack of dystrophin in these cells contribute to the temporal changes in activation status. There are several experiments that could

provide these data, e.g. BMT from WT mouse into mdx mice; showing that also the WT progenitor will show trained immunity.

Response: The reviewer has queried whether lack of dystrophin expression in myeloid cells per se could induce the characteristic features of trained immunity observed in mdx mice. However, in the mdx4cv mouse (with mutated exon 53) used in our study as well as in the original mdx mouse model (with mutated exon 23), the locations of these dystrophin gene mutations are upstream from the promoter of the short Dp71 dystrophin isoform expressed in myeloid cells^{7, 8}. Therefore, Dp71 expression is unaltered in these mdx mice⁹. This has recently been specifically confirmed in the mdx4cv spleen¹⁰. In the revised manuscript we have added this information to the Discussion section (first paragraph).

In addition, as noted by the reviewer, our original submission showed that pre-necrotic BMDM from mdx mice do not exhibit basal upregulation of prototypical M1/M2 genes or the characteristic abnormalities of mitochondrial oxygen consumption associated with trained immunity. We have now performed additional experiments which greatly strengthen these observations. In this regard, we find that during the pre-necrotic period there is a lack of significant differences between WT and mdx BMDM for both lactate levels (Figure 2d) and the M1/M2 gene transcript responses to in vitro stimulation (Supplemental Figure 3a-c).

2. To increase the impact of their findings, it would further strengthen the data if the authors could demonstrate markers of trained immunity also in human circulating monocytes from patients with DMD (e.g. increased cytokine production capacity, typical histone modifications).

Response: We agree entirely with the reviewer that this would be a powerful demonstration of the direct applicability of our findings to the human disease state. However, such a study would be a major undertaking. One would need to adequately account for factors such as age, disease stage, and medications (especially corticosteroid type and dosage), which would require a substantial number of control patients receiving corticosteroids as well as DMD patients. Unfortunately, we do not have ready access to such samples at this time, and to obtain these data would involve an extended time delay beyond the editorial limits of this manuscript submission. In the concluding paragraph of the revised manuscript, we have added the cautionary statement that “whether these phenomena also occur in human DMD or other forms of muscular dystrophy remains to be determined”.

3. The main read out for 'trained immunity' is the enhanced cytokine production capacity. The authors only measure mRNA expression throughout the manuscript. Please also provide data on extracellular cytokine protein concentrations.

Response: As suggested by the reviewer, we have now performed complementary western blot and ELISA experiments to confirm the mRNA findings at the protein level for several prototypical inflammatory mediators (iNOS, Arginase-1, TNF- α , IL-6). These new data are presented in Figure 5i-j of the revised manuscript.

MINOR COMMENTS:

1. Lines 60-68 describe the M1-M2 paradigm of macrophage polarization. First, I would recommend to describe that this is a highly simplified model and that there are multiple phenotypic states dependent on the specific micro-environment (Xue, Immunity 2014). Also, the authors use the stimuli that are classically used for M1/M2 differentiation to restimulate the cells for 24 hours. However, this is not differentiation, but restimulation of trained cells. Why do the authors use these stimuli?

Response: Thank you for raising these important points. Indeed, our own data also indicate the limitations of the M1/M2 paradigm since we found that both M1 and M2 prototype genes are simultaneously upregulated in the trained mdx BMDM in vitro. Accordingly, as suggested by the reviewer we have modified the statements in question to emphasize the oversimplification and inherent limits of the M1/M2 macrophage polarization paradigm, particularly when applied to the in vivo situation (see paragraph 3 of revised Introduction).

*The reviewer has also asked that we clarify our rationale for stimulating BMDM with cytokines which are able to induce M1 or M2 differentiation. In the revised manuscript, we have now explicitly stated that our intent was to use these agents as secondary stimuli and not as inducers of M1 or M2 differentiation. The basic objective was to determine if mdx BMDM show exaggerated responses to a wide variety of unrelated secondary stimuli (cytokines, fibrinogen, beta-glucan). This non-specific hyperresponsiveness is considered a cardinal feature of trained immunity. In other words, trained cells are expected to show an augmented response to multiple types of secondary stimuli even though the latter are not the same as the initial "training agent" stimulus and are also substantially different from one another. For example, vaccination against tuberculosis with BCG (the initial training stimulus) leads to stronger inflammatory responses against the unrelated secondary stimuli of *Candida albicans* (a fungus) and *Staphylococcus aureus* (a bacterium)¹¹. The cytokines IL-4 and IL-13 have also been used as*

secondary stimuli¹². In trained immunity, these non-specific and stronger “memory” responses to unrelated secondary stimuli are believed to result from the more open chromatin state conferred by epigenetic modifications. The augmented response to heterologous stimuli is thus considered a key feature of the trained immunophenotype, a concept which is nicely demonstrated in Figure 3 of an excellent recent review by Mihai Netea¹³.

In retrospect, our rationale for using the different secondary stimuli was perhaps not well explained in the original manuscript. We agree with the reviewer’s concern that there could be confusion about whether we were trying to induce classical M1 or M2 differentiation. Therefore, in the revised Results section under the heading “**mdx BMDM respond in an exaggerated fashion to heterologous inflammatory stimuli**”, in the opening sentences we have attempted to better explain the above rationale and more clearly indicate that our experimental design employed these agents as non-specific secondary stimuli in order to provide robust evidence for trained immunity in mdx BMDM.

2. Figure 2: please add what injections were given in the OCR-time graphs.

Response: Thank you for the suggestion, this has been done.

3. Fig 2: please also provide the lactate data for the pre-necrotic phase.

Response: The new experiment from the pre-necrotic period requested by the reviewer has now been done. As indicated earlier in our response to Major Comment #1, there were no differences in lactate levels between the WT and mdx groups (see revised Figure 2d).

4. Lines 140-142: the authors conclude that the major difference is acute versus chronic muscular injury. Another difference could be that the mechanism of muscle cell injury (and hence the DAMPS released) is completely different between the two models.

Response: We agree with the reviewer on this point and have added this comment to the revised Discussion section (last sentence of second paragraph).

5. Figure 3: the authors stimulate BMDMs with LPS/IFN or IL4 for 4 and 24 hours and describe M1- and M2 polarization in the result section. However, this is not a model for M1/M2 differentiation, but just restimulation of (trained/untrained) cells; 4 hours is too short for M1/M2 differentiation.

*Response: We agree entirely with the reviewer and as explained in the earlier response to Minor Comment #1, our goal was not to induce M1 or M2 differentiation. We believe that our data, demonstrating stronger responses to multiple unrelated secondary stimuli (cytokines, fibrinogen, beta-glucan), is powerful evidence for this defining feature of trained immunity. As noted earlier, a more explicit explanation for our rationale and experimental strategy is now provided in the revised Results section under the heading “**mdx BMDM respond in an exaggerated fashion to heterologous inflammatory stimuli**” in the opening sentences.*

6. Page 8, why did the authors restimulated with beta-glucan? In the context of trained immunity, this is a classical stimulus used to induce training, but this is never used as stimulus for restimulation. Please explain.

*Response: It is true that beta-glucan has often been used as a training stimulus. Indeed, the training stimulus effect of the fungus *Candida albicans* (another classical inducer of trained immunity) has been closely linked to its beta-glucan content¹⁴. However, *Candida albicans* has also been used as a secondary restimulation agent in several studies of trained immunity^{11, 14, 15}. As a matter of general principle, any pathogen-associated molecular pattern (PAMP) molecule has the potential to act as a secondary stimulus that can trigger exaggerated inflammatory responses in the presence of trained immunity. As noted earlier, we believe an important strength of our data is the demonstration that mdx BMDM show such exaggerated responses to multiple types of secondary stimuli, including but not limited to beta-glucan. As noted in the previous responses we have attempted to better clarify these points in the revised manuscript.*

7. Page 9: BMDMs are exposed to crushed skeletal muscle extracts. If the authors hypothesize that DAMPS from these muscles are released into the blood stream to affect bone marrow progenitors, it would also be possible to induce training by exposing BMDMs to serum from the mdx mice versus WT mice. This could also be done with serum from patients with DMD.

Response: We have performed the experiment suggested by the reviewer. Using the same design as the crushed muscle extract protocol shown in Figure 4a of the original manuscript, cultured BMDM from WT mice were exposed for 24 hours to PBS (control) or to sera from WT or mdx mice. The cells were then

washed and allowed to rest for 5 days, followed by restimulation with fibrinogen and collection of the cells. Both groups initially exposed to sera (WT or mdx) showed stronger secondary gene expression responses to fibrinogen compared to the PBS group, which is similar to the muscle extract model data. However, differences between the WT and mdx sera-stimulated groups were less pronounced than for muscle extract stimulation. These new data are presented in Supplemental Figure 4e.

We believe it is difficult to directly compare the muscle extract and serum stimulation experiments for several reasons. First, the absolute quantity of muscle-derived DAMPs is likely greater within crushed muscle extract. Second, the concentrations of other potential modifiers (eg., cytokines, exosomes) are likely quite different in the two models. Third, it is possible that the kinetics of the response differs. For example, a longer duration of stimulation (mimicking chronic serum exposure *in vivo*) might be required for mdx serum to exert its maximal effects on BMDM. To directly address these issues of dosage and exposure time, we performed additional experiments which are now presented in Supplemental Figure 4c-d. These studies indicate that induction of the trained immunophenotype in WT BMDM is dependent upon both the muscle extract concentration and its duration of exposure to the cells. In the revised manuscript, we now discuss these issues in more detail. In addition, we also acknowledge that chronic DAMP release from dystrophic muscles in mdx mice as the main stimulus for inducing trained immunity, while a very plausible explanation which is supported by our findings, cannot be absolutely proven to be the primary mechanism *in vivo* and thus remains a hypothesis (see paragraph 2 of revised Discussion section).

8. Why did the authors resstimulate with fibrinogen? This is an endogenous compound present in high concentrations in the circulation. Does this trigger cytokine release in immune cells?

Response: We selected fibrinogen based on the following criteria: 1) its longstanding recognition as an inducer of cytokine production by macrophages^{16, 17, 18}; 2) the fact that serum concentrations of fibrinogen are abnormally elevated in both mdx mice¹⁹ and in human DMD patients²⁰; and 3) the direct implication of fibrinogen in DMD pathogenesis, as shown by experiments in which either genetic or drug-induced deficiency of fibrinogen led to improved dystrophic pathology in mdx mice¹⁸. To address the reviewer's question we have expanded upon our rationale for performing secondary stimulation with fibrinogen in the revised manuscript (see paragraph 3 of Discussion).

9. On page 10, the authors describe the results of the mdxTLR4^{-/-} mice and show less inflammatory reprogramming of the BMDMs. Since TLR4 deficiency is not restricted to the myeloid lineage, I can imagine that TLR4^{-/-} also leads to a lower degree of muscular necrosis/inflammation/DAMP release. The authors should show that the stimulus (DAMPs) that allegedly reprograms the progenitor cells is unaffected in these mice.

Response: This is an interesting point. As suggested by the reviewer, we previously reported a reduced level of muscle pathology in mdx mice lacking TLR4⁶. It is thus possible that at least some DAMPs capable of inducing trained immunity are decreased in these mice. Unfortunately, it is impossible to directly prove or disprove this hypothesis in vivo, since we do not know which specific DAMPs are driving the generation of trained immunity in mdx BMDM and the number of potential candidates is vast²¹.

As an alternative strategy for addressing the reviewer's comment, we have performed additional in vitro experiments to determine whether the specific lack of TLR4 expression in BMDM is able to directly impair the development of trained immunity. Accordingly, using the same in vitro protocol which successfully trained WT BMDM after exposure to DAMPs contained in mdx muscle extract, we now demonstrate that BMDM from TLR4-deficient mice (non-dystrophic) do not develop the trained phenotype under the identical DAMP exposure conditions. This is evinced by the lack of hyperresponsiveness to secondary heterologous stimuli in the TLR4^{-/-} BMDM (shown in the updated Figure 5k-l of the revised manuscript). Therefore, these new data establish proof-of-concept that TLR4 expression in BMDM per se is required for the optimal induction of trained immunity by muscle DAMP exposure.

10. Page 11: please explain why you focused on K27me3, compared to other markers classically linked to trained immunity (K4me3, K4me, K27Ac).

Response: As indicated by the reviewer, most of the focus in prior studies of trained immunity has been on H3K27ac and H3K4me3. Therefore, we thought it would be of significant novelty and interest to initially focus on the less studied mechanism of H3K27 trimethylation. Of course, we agree that the other histone marks are also of considerable interest.

To address the reviewer's specific comments, we have now performed additional ChIP-Seq experiments to assess H3K27 acetylation. Overall, our data suggest a complex pattern of epigenetic reprogramming in mdx BMDM, with increases in histone mark modifications that can either augment or reduce the open chromatin state in different genes. We found that H3K27ac is reduced in mdx compared to WT, and even lower in mdxTLR4^{-/-} BMDM. These findings are consistent with previous work in trained immunity showing that methylation and acetylation

patterns can be simultaneously altered in a manner which has opposing effects on chromatin accessibility²². We speculate that this mixture of activating and repressive marks could serve to place the cells in a “poised” state able to rapidly adapt to different inflammatory environments (see paragraph 4 of Discussion section). The predominant histone mark changes demonstrated in mdxTLR4^{-/-} BMDM (increased H3K27me3 and decreased H3K27Ac), on the other hand, should both favor a decrease in open chromatin. In addition, among the epigenetic modifications observed in mdx BMDM, it was most clearly evident for H3K27me3 that the changes were both TLR4-dependent and predicted to promote increased transcriptional activation of genes involved in inflammation and fibrosis. Therefore, at least for H3K27, it appears that greater chromatin accessibility in mdx BMDM is mediated through reduced methylation rather than an increased level of acetylation. These new data are placed into the context of prior studies in the revised manuscript (see paragraphs 6 & 7 of Discussion section), and the new ChIP-Seq data for H3K27ac are outlined in the revised Results section as well as being presented in detail within Supplemental Figures 8 & 9.

Reviewer #3 (Remarks to the Author):

In this manuscript, the authors describe the involvement of trained immunity in the pathogenesis of Duchenne muscular dystrophy (DMD). Firstly, the authors utilized bone marrow derived macrophages (BMDMs) and conducted a series of real time PCR assays, They found that, in the mdx mice, which is a model of DMD, authentic features of trained immunity was observed. In the mdx mice, trained immunity was hypersensitive compared to the wild type mice. Those responses were invoked by muscle extract, suggesting its important role in etiology of DMD. They also examined and found that all the molecular responses and phenotypic appearance were TLR4 dependent. Importantly, the authors conducted an in vivo experiment and found that the hypersensitive phenotype was transmissible by bone marrow transplantation to healthy mice. The authors also describe that the training of this innate immune response should be mediated by the epigenetic modifications of particular genomic regions in BMDMs.

Overall, I fully agree that this paper precisely aiming at the molecular features underlying this difficult disease. In fact, the training mechanisms of the immune system in DMD remains almost totally unknown and its better understandings may open a new research field for drug development or designing other therapeutic strategies. However, after reading through this manuscript, I still have remaining concern what is the essence of the molecular memory of this phenotype of the mdx mice. The presented data is mostly from the rather straightforward gene expression analyses and the complex or heterogeneous

nature of its association with epigenome elements are still remaining uncharacterized.

Response: We agree with the reviewer that determination of the precise molecular basis for innate immune memory is a major challenge. Previous investigations in the field of trained immunity have focused on a limited number of epigenetic modifications. Therefore, even for classical inducers of trained immunity such as BCG or Candida albicans, this question has not been answered.

In consideration of the reviewer's comments, we have significantly expanded the scope of our epigenetic analysis by performing additional ChIP-Seq experiments to assess H3K27 acetylation. These new data are placed into the context of prior studies in the revised manuscript (see paragraphs 6 & 7 of Discussion section), and the new ChIP-Seq data for H3K27ac are outlined in the revised Results section as well as being presented in detail within Supplemental Figures 8 & 9.

Overall, our data suggest a complex pattern of epigenetic reprogramming in mdx BMDM, with increases in histone mark modifications that can either augment or reduce the open chromatin state in different genes. These findings are consistent with previous work in trained immunity showing that methylation and acetylation patterns can be simultaneously altered in a manner which has opposing effects on chromatin accessibility²². We speculate that this mixture of activating and repressive marks could serve to place the cells in a "poised" state able to rapidly adapt to different inflammatory environments (see paragraph 4 of Discussion section). The predominant histone mark changes demonstrated in mdxTLR4^{-/-} BMDM (increased H3K27me3 and decreased H3K27Ac), on the other hand, should both favor a decrease in open chromatin. In addition, among the epigenetic modifications observed in mdx BMDM, it was most clearly evident for H3K27me3 that the changes were both TLR4-dependent and predicted to promote increased transcriptional activation of genes involved in inflammation and fibrosis. Therefore, at least for H3K27, it appears that greater chromatin accessibility in mdx BMDM is mediated through reduced methylation rather than an increased level of acetylation.

We believe that these observations are highly original in the muscular dystrophy field and provide valuable new insights into the behavior of macrophages in DMD. However, we fully recognize that this is very likely a small representation of the number of epigenetic changes which take place in mdx BMDM, and have made a specific comment on this point in the revised manuscript (paragraph 7 of the Discussion).

Major points:

1. It is essential to further elucidate the epigenomic features of BMDMs. Indeed,

Figure 6e clearly represents the “changes” in epigenomic features in the wt. mdx and mdx/TLR4(-/-) mice. However, this data also indicates that there should be certain cellular heterogeneity within the population. Considering there are only two copies of genomic DNA per cell, the observed changes should represent the changes in the cellular populations having the corresponding epigenomic patterns. This heterogeneity may be further associated with the separation of BMDM potentials in pro-inflammatory or anti-inflammatory milieus. To further elucidate these features, I consider the single cell analysis should be essential. Particularly, the simultaneous measurement of the gene expressions and the open chromatin structures of a give cell has become rapidly easy and popular.

Response: We agree with the reviewer that there might be heterogeneity in the mdx BMDM population with regard to training effects, with different subpopulations of BMDM showing different gene expression and epigenomic patterns. Therefore, the single cell multiomics measurements that the reviewer suggests could certainly provide a more refined understanding of the trained immunity phenomenon in mdx mice. Indeed, it is an area that we would like to explore in a future study.

However, the main objective of the present study was to first establish whether trained immunity exists in DMD macrophages as a pathological phenomenon. To support this goal, we wanted to perform a systematic exploration of epigenetic patterns associated with all genes across the entire genome. There are a few reasons that make the bulk-level measurements a more suitable choice here.

Although single cell analysis can deconvolve the cell heterogeneity, it may also miss many critical genes that could otherwise be captured by the bulk-level approach. The read depths for single cell measurements are much shallower compared to the bulk measurement, thus limiting our ability to investigate the epigenetic patterns associated with all genes across the whole genome. Hence the bulk measurements capture more genes compared to the single cell counterparts. For example, from the H3K27me3 bulk measurement we have identified 2293 genes with GP1 (IUI) pattern, 786 genes with GP1 (IUU) pattern, 154 genes with GP3 (UII) pattern, and 66 genes with GP4 (UIU), for a total of 3299 dynamically regulated epigenetic patterns in the mdx group. In contrast, the mean number of detected genes from the most available 10x Genomics single cell RNA-seq platform is around 2000 genes²³. Thus, the single cell approach would have missed a significant proportion of genes with dynamically regulated epigenomic patterns which we detected in the mdx group (3299 captured by our bulk data). In fact, the current single cell epigenetic measurements such as single cell ATAC-seq or single cell methylation analyses are noisier and more sparse in read depth than single cell RNA-seq²⁴. Furthermore, because of previous studies indicating that histone modifications are important in the induction of trained immunity, we logically focused on this aspect of epigenetic

regulation in our study. Unfortunately, single cell histone modification measurements (eg. scCHIC-seq) are not yet readily available and therefore our analysis of histone marks could not have been performed using single cell technology.

2. Also, the status changes and the timing of the establishment for the immune cell training should be further clarified. For example, in the mdx mice, would not pre-necrotic BMDM show any responses to the stimulation?

Response: We have performed additional experiments as suggested by the reviewer. We find that during the pre-necrotic period there is a lack of significant differences between WT and mdx BMDM for both lactate levels (Figure 2d) and the M1/M2 gene transcript responses to in vitro stimulation (Supplemental Figure 3a-c). Therefore, these data greatly strengthen our observations in the original manuscript, which showed that pre-necrotic BMDM from mdx mice do not exhibit basal upregulation of prototypical M1/M2 genes or the characteristic abnormalities of mitochondrial oxygen consumption associated with trained immunity.

3. Phenotypic appearances of the mice, such as the degree of the muscle destruction, should be more quantitatively analyzed. From a clinical viewpoint, it is essential to learn to what extent trained immunity is involved in the pathology of DMD. Such an information is particularly important, when the development of a new drugs or other therapeutic strategy should be considered in this direction.

Response: Thank you for this suggestion. In a previous publication, we performed a detailed analysis showing that skeletal muscle pathology is less severe in mdx mice lacking TLR4, and that the improved histology is also associated with greater force-generating capacity of the main respiratory muscle, the diaphragm⁶. In the revised manuscript we have now placed this prior report and the new findings of this paper into greater context with respect to their potential therapeutic implications (paragraph 3 and final paragraph of Discussion section).

Minor point:

4. It is essential to further analyze how TLR4 should transduce the signal to the downstream molecules. Some pathway analyses are described but no validation is conducted. Also, it should be validated how the mRNA changes may lead to

the protein level inductions for the examined pro-inflammatory, anti-inflammatory and other molecules.

Response: As suggested by the reviewer, we have now performed complementary western blot and ELISA experiments to confirm the mRNA findings at the protein level for several prototypical inflammatory mediators (iNOS, Arginase-1, TNF- α , IL-6). These new data are presented in Figure 5i-j of the revised manuscript. We agree that elucidation of the TLR4 signaling pathways which drive epigenetic changes is of considerable interest, but we believe this is a very complex area which should be the topic of a separate study.

5. It is remaining elusive, at least to me, how the innate immune system may have acquired a specificity, if any, to the muscle extract, assuming there should be other antigens, particularly when we think about the case in human patients.

Response: Thank you for this comment. We agree that there is no specificity of the innate immune system to muscle extract. This is demonstrated by the fact that WT BMDM exposed to muscle extract in vitro as well as mdx BMDM exposed to muscle necrosis in vivo respond in an exaggerated fashion to other secondary stimuli. This non-specific hyperresponsiveness is considered a typical feature of trained immunity. For example, vaccination against tuberculosis with BCG (the initial training stimulus) leads to stronger inflammatory responses against the unrelated secondary stimuli of *Candida albicans* (a fungus) and *Staphylococcus aureus* (a bacterium)¹¹. In trained immunity, these non-specific and stronger “memory” responses are believed to result from the more open chromatin state conferred by epigenetic modifications. In retrospect, we realize that this concept was not adequately explained in the original manuscript and we have attempted to improve our explanation in the revised version (for example, in the Results section under the heading “**mdx BMDM respond in an exaggerated fashion to heterologous inflammatory stimuli**”).

6. Extensive analyses is needed to further inspect the changes of H3K4me3 patterns. Particularly, I wonder whether the H3K4me3 pattern did not change around the transcriptionally activated genes or the activated genes were only minor. If the former was the case, what is the suspected controlling factor?

Response: As indicated in our first response to the reviewer’s general comments, we have performed additional extensive analyses of H3K27ac to expand the scope of our epigenetic analyses. We decided to focus on H3K27ac for the following reasons: 1) this is the histone mark most frequently linked to trained immunity in previous studies of infection models; and 2) this allowed us to directly contrast findings at the same histone location (ie., H3K27) for acetylation and

trimethylation status. Our data for H3K4me3 are more limited in scope but the ChIP-PCR analyses are consistent with greater promoter occupancy for the genes showing higher mRNA transcript levels. However, it is also clear that the situation is extremely complex with many potential epigenomic modifications taking place at the same time, and this point is now emphasized in the revised manuscript (paragraphs 4 and 7 of revised Discussion).

7. In fact, trained immunity has been indicated to be involved in other chronic diseases and cancers (Netea et al Nature Reviews Immunology 2020) in addition to DMD. Discussion should be enriched from such a viewpoint.

Response: Thank you for this suggestion. We have emphasized the chronic disease relevance of trained immunity (paragraph 4 of Introduction and concluding paragraph of the Discussion).

8. p28: Data analysis of ChIP seq should be conducted in a more quantitative manner.

Response: To address the reviewer's comment, we now provide more detail about the quantitative methods used for the ChIP-seq analyses within the revised Methods section.

9. Title is somewhat ambiguous to represent the contents of the manuscript.

Response: We have modified the title of the paper to specify that the study was performed in muscular dystrophy mice.

10. If any of the experimental procedures was hinted by a previous paper, please refer to it (for example, de Laval et al, Cell Stem Cell, 2020?).

Response: Thank you for pointing out this interesting paper which is indeed very relevant to our study. We now refer to this paper in the Introduction, Results and Discussion sections of the revised manuscript.

References

1. Matecki, S., Guibinga, G.H. & Petrof, B.J. Regenerative capacity of the dystrophic (mdx) diaphragm after induced injury. *Am J Physiol Regul Integr Comp Physiol* **287**, R961-968 (2004).
2. Pessina, P. *et al.* Novel and optimized strategies for inducing fibrosis in vivo: focus on Duchenne Muscular Dystrophy. *Skelet Muscle* **4**, 7 (2014).
3. Desguerre, I. *et al.* A new model of experimental fibrosis in hindlimb skeletal muscle of adult mdx mouse mimicking muscular dystrophy. *Muscle Nerve* **45**, 803-814 (2012).
4. Bekkering, S. *et al.* In Vitro Experimental Model of Trained Innate Immunity in Human Primary Monocytes. *Clin Vaccine Immunol* **23**, 926-933 (2016).
5. Nielsen, F. Introduction to HPC with MPI for Data Science. *Undergraduate Topics in Computer Science*,. 1st ed. Cham: Springer International Publishing : Imprint: Springer,; 2016.
6. Giordano, C. *et al.* Toll-like receptor 4 ablation in mdx mice reveals innate immunity as a therapeutic target in Duchenne muscular dystrophy. *Hum Mol Genet* **24**, 2147-2162 (2015).
7. Cerecedo, D., Cisneros, B., Gomez, P. & Galvan, I.J. Distribution of dystrophin- and utrophin-associated protein complexes during activation of human neutrophils. *Exp Hematol* **38**, 618-628 e613 (2010).
8. Tadayoni, R., Rendon, A., Soria-Jasso, L.E. & Cisneros, B. Dystrophin Dp71: the smallest but multifunctional product of the Duchenne muscular dystrophy gene. *Mol Neurobiol* **45**, 43-60 (2012).
9. Im, W.B. *et al.* Differential expression of dystrophin isoforms in strains of mdx mice with different mutations. *Hum Mol Genet* **5**, 1149-1153 (1996).
10. Dowling, P. *et al.* Proteome-wide Changes in the mdx-4cv Spleen due to Pathophysiological Cross Talk with Dystrophin-Deficient Skeletal Muscle. *iScience* **23**, 101500 (2020).

11. Kleinnijenhuis, J. *et al.* Bacille Calmette-Guerin induces NOD2-dependent nonspecific protection from reinfection via epigenetic reprogramming of monocytes. *Proc Natl Acad Sci U S A* **109**, 17537-17542 (2012).
12. Ordovas-Montanes, J. *et al.* Allergic inflammatory memory in human respiratory epithelial progenitor cells. *Nature* **560**, 649-654 (2018).
13. Netea, M.G., Schlitzer, A., Placek, K., Joosten, L.A.B. & Schultze, J.L. Innate and Adaptive Immune Memory: an Evolutionary Continuum in the Host's Response to Pathogens. *Cell Host Microbe* **25**, 13-26 (2019).
14. Quintin, J. *et al.* Candida albicans infection affords protection against reinfection via functional reprogramming of monocytes. *Cell Host Microbe* **12**, 223-232 (2012).
15. Rizzetto, L. *et al.* Fungal Chitin Induces Trained Immunity in Human Monocytes during Cross-talk of the Host with *Saccharomyces cerevisiae*. *J Biol Chem* **291**, 7961-7972 (2016).
16. Sitrin, R.G., Pan, P.M., Srikanth, S. & Todd, R.F., 3rd. Fibrinogen activates NF-kappa B transcription factors in mononuclear phagocytes. *J Immunol* **161**, 1462-1470 (1998).
17. Smiley, S.T., King, J.A. & Hancock, W.W. Fibrinogen stimulates macrophage chemokine secretion through toll-like receptor 4. *J Immunol* **167**, 2887-2894 (2001).
18. Vidal, B. *et al.* Fibrinogen drives dystrophic muscle fibrosis via a TGFbeta/alternative macrophage activation pathway. *Genes Dev* **22**, 1747-1752 (2008).
19. Hathout, Y. *et al.* Discovery of serum protein biomarkers in the mdx mouse model and cross-species comparison to Duchenne muscular dystrophy patients. *Hum Mol Genet* **23**, 6458-6469 (2014).
20. Hathout, Y. *et al.* Large-scale serum protein biomarker discovery in Duchenne muscular dystrophy. *Proc Natl Acad Sci U S A* **112**, 7153-7158 (2015).

21. Gong, T., Liu, L., Jiang, W. & Zhou, R. DAMP-sensing receptors in sterile inflammation and inflammatory diseases. *Nat Rev Immunol* **20**, 95-112 (2020).
22. Bekkering, S. *et al.* Innate immune cell activation and epigenetic remodeling in symptomatic and asymptomatic atherosclerosis in humans in vivo. *Atherosclerosis* **254**, 228-236 (2016).
23. Wang, X., He, Y., Zhang, Q., Ren, X. & Zhang, Z. Direct Comparative Analyses of 10X Genomics Chromium and Smart-seq2. *Genomics Proteomics Bioinformatics* (2021).
24. Chen, H. *et al.* Assessment of computational methods for the analysis of single-cell ATAC-seq data. *Genome Biol* **20**, 241 (2019).

REVIEWER COMMENTS

Reviewer #1 (Remarks to the Author):

The authors did a good job addressing the issues raised. However, I'm still puzzled by the potential implications of the findings in the context of chronically inflammatory dystrophic muscles in Duchenne patients. It appears that the trained immunity occurred primarily at the necrotic phase which is unique to mdx mice and does not exist in human patients. The characteristic features of trained immunity are greatly attenuated in 1-year-old mdx mice (Figures 1c and 2c). I'd consider chronic inflammation a feature of 1-year-old mdx mice. Muscle inflammation in 6-week-old mdx mice should be considered as acute or sub-acute. With this said, I'd expect BMDM from 1-yr-old mdx to show more pronounced responsiveness to secondary non-specific stimuli. I'm wondering whether the authors have done this experiment (I cannot find it, but it is possible that I may have missed it).

Reviewer #2 (Remarks to the Author):

The authors have been able to address all my concerns and suggestions. They have performed several additional experiments (measurement of protein levels in addition to mRNA expression; training with serum of WT/mdx mice; training of BMDM from TLR4^{-/-} mice with mdx muscle extract; additional ChIP-seq for H3K27Ac), which significantly strengthen the manuscript.

Unfortunately, they were not able to perform the proposed experiments to phenotype circulating monocytes of patients with DMD, since this would require too much time and effort. I can understand that and appreciate the addition in the revised version that translation of their findings to the patient setting remains to be determined.

I have two small remaining remarks on the new H3K27Ac results. On page 14, lines 285, the authors state that '...expected to decrease chromatin accessibility'. I don't understand that statement, since the decrease of the (repressive) H3K27me3 mark in mdx BMDM would fit with increased chromatin accessibility. This is also true for the enrichment of H3K4me3. Please explain or amend.

In the discussion on page 21, line 433, the authors state that ref 49 shows opposing alterations in acetylation and methylation. However, ref 49 only studied methylation; they showed a reduction in H3K4me3, but also a reduction in the repressive mark H3K27me3, just like the authors in the current paper. So, I would suggest to change this into '...a previous report showing a simultaneous decrease in activating and repressive methylation marks'.

Reviewer #3 (Remarks to the Author):

First of all, I appreciate the authors' substantial efforts for the revision. In fact, thanks to their dedicated efforts, the contents of the manuscript have been much enriched. The technical concerns which I raised in the previous round of the review have been mostly properly addressed. Honestly, I still have a remaining concern about the cellular diversity underlying the observed phenomena. However, I believe further continuous efforts of the authors would make the obtained knowledge useful for clinical needs. Overall, I consider that the publication of this paper should have been fully rationalized now.

The authors express their profound thanks to all of the reviewers for their thoughtful and constructive comments, which have clearly served to substantially strengthen the manuscript.

Reviewer #1 (Remarks to the Author):

The authors did a good job addressing the issues raised. However, I'm still puzzled by the potential implications of the findings in the context of chronically inflammatory dystrophic muscles in Duchenne patients. It appears that the trained immunity occurred primarily at the necrotic phase which is unique to mdx mice and does not exist in human patients. The characteristic features of trained immunity are greatly attenuated in 1-year-old mdx mice (Figures 1c and 2c). I'd consider chronic inflammation a feature of 1-year-old mdx mice. Muscle inflammation in 6-week-old mdx mice should be considered as acute or sub-acute. With this said, I'd expect BMDM from 1-yr-old mdx to show more pronounced responsiveness to secondary non-specific stimuli. I'm wondering whether the authors have done this experiment (I cannot find it, but it is possible that I may have missed it).

Response: The data requested by the reviewer, showing mdx BMDM hyperresponsiveness to secondary non-specific stimuli during the fibrotic stage of disease, were provided in Supplementary Fig 2 of the last submission. These results from the fibrotic stage were described in the text of the manuscript on page 8 where we stated: "An analogous but less pronounced pattern of generalized hyperresponsiveness to these stimuli was also found in mdx BMDM from the fibrotic phase of the disease (Supplementary Fig. 2)". We acknowledge the reviewer's point regarding differences in the degree of necrosis between the mdx mouse model and human DMD, and have now added a statement about this difference in the newly revised manuscript (last paragraph of the Discussion section).

Reviewer #2 (Remarks to the Author):

The authors have been able to address all my concerns and suggestions. They have performed several additional experiments (measurement of protein levels in addition to mRNA expression; training with serum of WT/mdx mice; training of BMDM from TLR4^{-/-} mice with mdx muscle extract; additional ChIP-seq for H3K27Ac), which significantly strengthen the manuscript.

Unfortunately, they were not able to perform the proposed experiments to phenotype circulating monocytes of patients with DMD, since this would require too much time and effort. I can understand that and appreciate the addition in the revised version that translation of their findings to the patient setting remains to be determined.

I have two small remaining remarks on the new H3K27Ac results. On page 14, lines 285, the authors state that '...expected to decrease chromatin accessibility'. I don't understand that statement, since the decrease of the (repressive) H3K27me3 mark in mdx BMDM would fit with increased chromatin accessibility. This is also true for the enrichment of H3K4me3. Please explain or amend.

In the discussion on page 21, line 433, the authors state that ref 49 shows opposing alterations in acetylation and methylation. However, ref 49 only studied methylation; they showed a reduction in H3K4me3, but also a reduction in the repressive mark H3K27me3, just like the authors in the current paper. So, I would suggest to change this into '...a previous report showing a simultaneous decrease in activating and repressive methylation marks'.

Response: Regarding page 14, line 285 the original text was as follows: “these data suggest a complex pattern of epigenetic reprogramming in mdx BMDM, with co-existence of histone mark modifications that can either augment or reduce the open chromatin state. The predominant histone mark changes demonstrated in mdxTLR4^{-/-} BMDM, on the other hand, would generally be expected to decrease chromatin accessibility”. Therefore, the statement in question (“expected to decrease chromatin accessibility”) does not refer to mdx BMDM but instead to the mdxTLR4^{-/-} BMDM group. We have amended the paragraph to more clearly spell out which histone mark changes apply to which BMDM group.

Regarding page 21, line 433 and its relationship to reference 49: The reviewer is absolutely correct and we greatly appreciate the care taken to point out this imprecision in the original text. We have corrected this statement as suggested.

Reviewer #3 (Remarks to the Author):

First of all, I appreciate the authors' substantial efforts for the revision. In fact, thanks to their dedicated efforts, the contents of the manuscript have been much enriched. The technical concerns which I raised in the previous round of the review have been mostly properly addressed. Honestly, I still have a remaining concern about the cellular diversity underlying the observed phenomena. However, I believe further continuous efforts of the authors would make the obtained knowledge useful for clinical needs. Overall, I consider that the publication of this paper should have been fully rationalized now.

Response: We thank the reviewer for all of the thoughtful suggestions, which have definitely improved the manuscript.

REVIEWER COMMENTS

Reviewer #1 (Remarks to the Author):

I have no further concerns.

REVIEWERS' COMMENTS

Reviewer #1 (Remarks to the Author):

I have no further concerns.

Response: We thank the reviewers for their constructive comments which have resulted in a better manuscript.